# Nonlinear feedback modulation contributes to the optimization of flexible decision-making

Xuanyu Wu[1], Yang Zhou[1,2,3,4]*

[1]Peking-Tsinghua Center for Life Sciences, Peking University, Beijing, China; [2]School of Psychological and Cognitive Sciences, Peking University, Beijing, China; [3]PKU-IDG/McGovern Institute for Brain Research, Peking University, Beijing, China; [4]Department of Neurobiology, The University of Chicago, Chicago, United States

## eLife Assessment

This **valuable** study by Wu and Zhou combines neurophysiological recordings and computational modelling to address an interesting question regarding the sequence of events from sensing to action. Neurophysiological evidence remains **incomplete**: explicit mapping of saccade-related activity in the same neurons and a better understanding of the influence of the spatial configuration of stimulus and targets would be required to pinpoint whether such activity might contribute, even partially, to the observed results and interpretations. These results are of interest for neuroscientists investigating decision-making.

*For correspondence:
yangzhou1@pku.edu.cn

Competing interest: The authors declare that no competing interests exist.

**Abstract** Neural activity in the primate brain correlates with both sensory evaluation and action selection aspects of decision-making. However, the intricate interaction between these distinct neural processes and their impact on decision behaviors remains unexplored. Here, we examined the interplay of these decision processes in posterior parietal cortex (PPC) when monkeys performed a flexible decision task. We found that the PPC activity related to monkeys' abstract decisions about visual stimuli was nonlinearly modulated by monkeys' following saccade choices directed outside each neuron's response field. Recurrent neural network modeling indicated that the feedback connections, matching the learned stimuli-response associations during the task, might mediate such feedback modulation. Further analysis on network dynamics revealed that selectivity-specific feedback connectivity intensified the attractor basins of population activity underlying saccade choices, thereby increasing the reliability of flexible decisions. These results highlight an iterative computation between different decision processes, mediated primarily by precise feedback connectivity, contributing to the optimization of flexible decision-making.

## Introduction

Perceptual decisions typically involve the transformation of sensory inputs to discrete motor responses. Consequently, sensory evaluation and action selection emerge as two fundamental processes essential for implementing perceptual decision behavior (*O'Connell et al., 2018*; *Freedman and Assad, 2016*; *Gold and Shadlen, 2007*; *Huk et al., 2017*; *Hanks and Summerfield, 2017*; *Shadlen and Kiani, 2013*; *Freedman and Assad, 2011*). Neural activity associated with either the sensory or motor aspects of decision-making is widely distributed across different brain areas, such as PPC (*Platt and Glimcher, 1999*; *Roitman and Shadlen, 2002*; *Shadlen and Newsome, 1996*; *Shadlen and Newsome, 2001*; *Sugrue et al., 2004*; *Yates et al., 2017*; *Zhou and Freedman, 2019*; *Freedman and Assad, 2006*;

*Freedman and Assad, 2009*; *Bennur and Gold, 2011*; *Swaminathan and Freedman, 2012*; *Zhou et al., 2022*; *Kiani and Shadlen, 2009*; *Katz et al., 2016*; *Zhou et al., 2023*), frontal cortex (*Swaminathan and Freedman, 2012*; *Kim and Shadlen, 1999*; *Rossi-Pool et al., 2017*; *Ding and Gold, 2012*; *Ferrera et al., 2009*; *Heitz and Schall, 2012*; *Zhou et al., 2021*), superior colliculus (*Horwitz and Newsome, 1999*; *Peysakhovich et al., 2023*), and caudate (*Ding and Gold, 2010*). A recent inactivation study has shown that the lateral intraparietal area (LIP), a subregion of PPC, plays a causal role in flexible visuomotor decisions, with preferential involvement in sensory evaluation rather than action selection (*Zhou and Freedman, 2019*). Meanwhile, areas outside PPC, like frontal eye field and superior colliculus, have been shown to be causally involved in the action selection aspect of decision-making (*Gold and Shadlen, 2000*; *Jun et al., 2021*). These findings suggest that sensory evaluation and action selection likely involve distinct neural processing in the primate brain during flexible decision-making.

However, previous reports have also shown that neural activity related to sensory evaluation and action selection overlaps temporally and spatially in the brain (*Shushruth et al., 2018*). This leaves open the possibility of interplay between these distinct processes during decision-making. From the perspective of information processing, the abstract result of sensory evaluation should be transmitted to the motor planning circuits to guide the action selection process in a feedforward manner, although both processes could proceed in parallel in the brain. However, whether and how action selection processes might exert a feedback influence on sensory evaluation have not been explicitly studied.

In this study, we examined the activities of single LIP neurons during a flexible decision task in which monkeys needed to report their decisions about a motion stimulus with a saccade to one of two color targets. Specifically, we arranged the motion stimuli inside each neuron's response field (RF) and positioned the saccade targets in the direction perpendicular to the axis of neural RFs. We found that LIP activity responding to visual motion was nonlinearly modulated by the monkeys' following saccade choice direction relative to the recorded brain hemisphere. Notably, this modulation was aligned precisely with the functional properties of each neuron, as well as specifically impacted the decision-correlated but not the stimulus-correlated activity of LIP neurons. This suggests a precise 'feedback' modulation from action selection process to the neural processing of sensory evaluation during decision-making.

RNN models trained on complex behavioral tasks have shown promise for understanding neural computations and circuit mechanisms underlying cognitive functions (*Engel et al., 2015*; *Masse et al., 2019*; *Song et al., 2016*). To further explore how action selection can modulate sensory evaluation during decision-making, we trained multi-module RNNs, which consist of different hemispheres and RF structures, to perform the same decision task and analyzed the activity of network units during task performance. These RNNs exhibited similar behavioral performance and neural activity patterns as observed in the monkey electrophysiology experiment, including patterns of nonlinear feedback modulation from action selection to sensory evaluation. Combining analysis on network activity and connectivity with projection-specific inactivation experiments in the RNNs, we further showed that the precise feedback connections between units that showed matched functional properties within different network modules were the key circuit mechanism for mediating the modulation of sensory evaluation by action selection. Such feedback modulation significantly enhanced the consistency of the RNNs' decisions by strengthening the attractor basins of network dynamics underlying saccade choices.

## Results

### Nonlinear modulation of saccade choice on visual motion selectivity in LIP

To test the mechanisms underlying the feedback modulation of action selection on sensory evaluation, we trained two monkeys to perform a reaction-time version of the flexible visual-motion discrimination (FVMD) task, in which they needed to choose one of two colored saccade targets based on their decisions about the motion direction of a sample stimulus shown at different coherence levels (*Figure 1A*). Monkeys learned the mappings between the two motion directions (315° and 135°) and two target colors (red and green) at the start of training, and the mappings were fixed across the study. Because the locations of the red and green targets were randomly interleaved across trials,

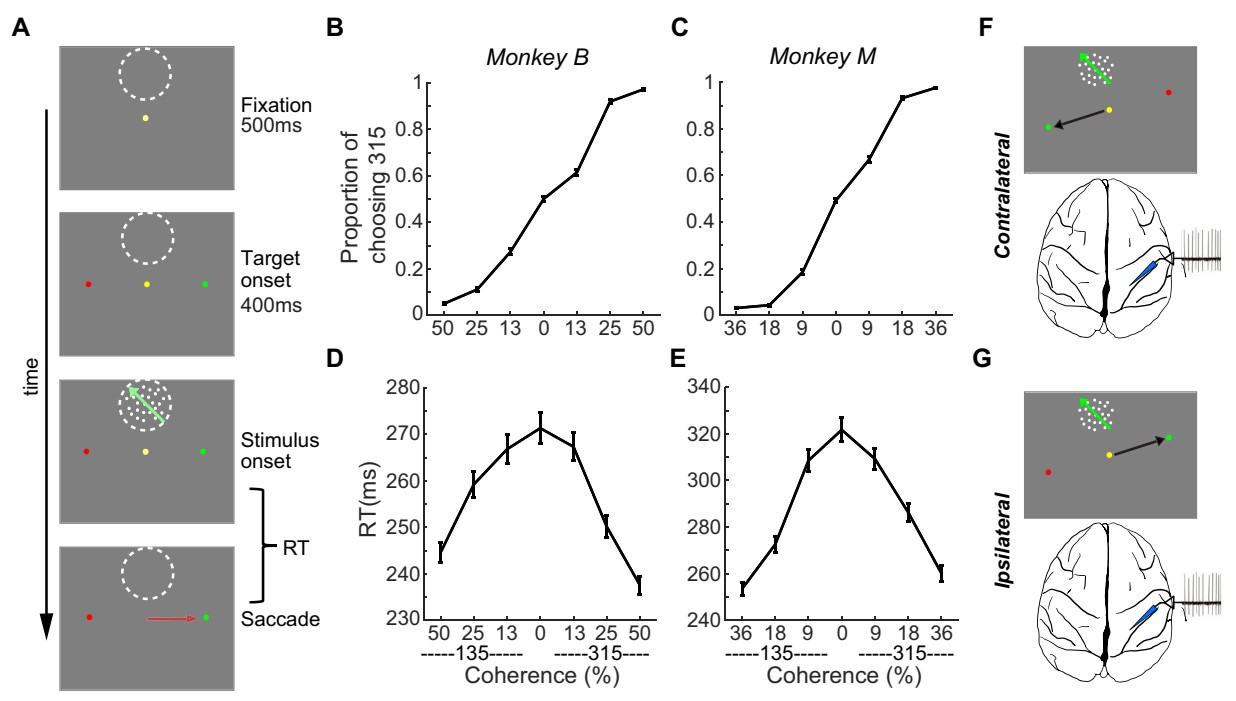

**Figure 1.** Behavioral task. (**A**) The flexible visual-motion discrimination (FVMD) task. Monkeys needed to report their decision about the direction of the visual motion stimuli by choosing either the green or red saccade target. The appearance of the two color targets preceded that of the visual-motion stimulus, and the target positions were randomly chosen on each trial to avoid fixed mapping between motion stimulus and saccade direction. Monkeys could initiate their saccade as soon as they had made their decision. (**B–C**) Psychometric curves for the two monkeys. The averaged performance accuracy from all recording sessions (N=125) for each monkey is plotted as the proportion of trials in which 315° was chosen as a function of the directions and coherence levels of the motion stimuli. Error bar denote ± SEM. (**D–E**) Chronometric curves are shown separately for the two monkeys. (**F–G**) Two trial conditions, contralateral (**F**) and ipsilateral (**G**), defined according to the spatial configurations of task stimuli during neural recording.

neither of the motion directions was directly linked with one specific saccade direction. Both the performance accuracy and the reaction time (RT) changed systematically as a function of motion coherence levels (*Figure 1B–E*): as the motion coherence increased, both monkeys chose the correct target more frequently and more rapidly.

We recorded single-neuron activity from the LIP while the monkeys performed the FVMD task. We used the memory-guided saccade (MGS) task, which is commonly employed in LIP studies, to map the receptive fields (RFs) of all isolated LIP neurons. Specifically, we mapped both the visual and memory RFs of each neuron by analyzing their activity during the target presentation and delay periods of the MGS task (see Materials and methods). To examine the neural activity related to the evaluation of stimulus motion, we presented the motion stimuli within the RF of each neuron, while positioning the saccade targets at locations orthogonal to the line connecting the center of the RF (which also marks the center of the motion stimulus) and the fixation dot. In total, 104 of 194 visually responsive LIP neurons (monkey M: 50/78; monkey B: 54/116) showed significant direction selectivity (DS) to the motion stimuli (one-way ANOVA, p<0.01). To examine the influence of action selection on sensory evaluation, we analyzed data from the subset of sessions in which the saccade targets were aligned more closely with the horizontal direction than the vertical direction (83 of 104 neurons). In these sessions, the motor planning corresponding to a saccade to either target would be mediated primarily by one brain hemisphere. We therefore defined the conditions under which the correct target was contralateral or ipsilateral to the recorded hemisphere as the contralateral target (CT) condition or ipsilateral target (IT) condition, respectively (*Figure 1F and G*).

*Figure 2A–F* shows three example LIP neurons that exhibited significant motion coherence correlated DS. Surprisingly, LIP neurons showed greater DS in the CT condition than in the IT condition, even though the same motion stimuli were used in the same spatial location for both conditions. The averaged population activity showed this DS difference between CT and IT conditions for all four coherence levels (*Figure 2G, H*). During the presentation of their preferred motion direction,

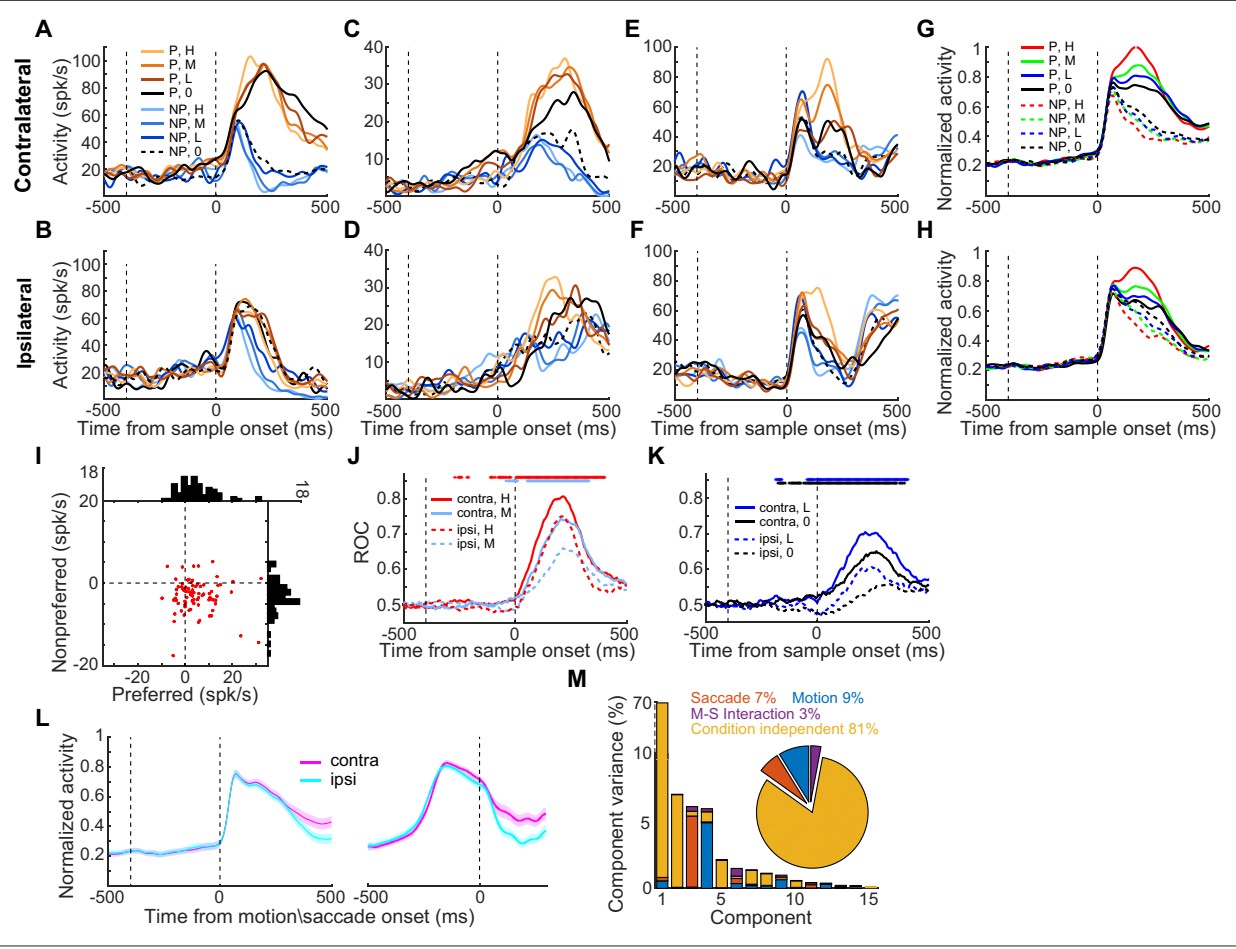

**Figure 2.** Nonlinear feedback modulation of saccade choice on sensory evaluation in LIP. (**A**) The activity of one example neuron in the CT condition of the FVMD task is shown for each motion-coherence level. The zero-coherence trials were grouped based on the monkey's choices. The two vertical dashed lines denote the time of target and motion stimulus onset, respectively. (**B**) The same example neuron's activity in the IT condition of the FVMD task. (**C–F**) The activities of two more example neurons. (**G–H**) The averaged population activities of all direction-selective neurons (N=83) that were collected during the recording sessions in which the saccade targets were arranged in either horizontal or oblique directions. The activity to each motion direction and coherence level is shown separately for the CT condition (**G**) and IT condition (**H**). (**I**) The activity differences between CT and IT conditions (CT minus IT) of single LIP neurons were plotted for both preferred and nonpreferred motion directions. Each dot represents the activity of a single neuron. The histograms in the horizontal and vertical axes represent the distribution of activity difference between CT and IT conditions for preferred and nonpreferred motion directions, respectively. (**J**) An ROC analysis was used for quantifying the motion DS for both CT (solid) and IT (dashed) conditions. The colored dots denote the time points for which there was a significant difference between the CT and IT conditions (paired t-test: p<0.01). (**K**) The average DS for low- and zero-coherence trials is shown as in (**J**). (**L**) The mean activity (averaged across all motion directions and coherence levels, shaded area denotes ± SEM) was compared between the two saccade directions (CT vs. IT) at the population level. Activity was aligned to either motion stimulus onset (left panel) or saccade onset (right panel). There was no significant difference between CT and IT conditions before monkeys made saccade choices. (**M**) Variance in LIP population activity as explained by the individual demixed principal components. Each bar shows the proportion of total explained variance that was contributed by the four task variables. The pie chart shows the total variance explained by each task variable. H, high; M, medium; L, low; 0, zero; P, preferred; NP, nonpreferred.

The online version of this article includes the following figure supplement(s) for figure 2:

**Figure supplement 1.** The comparisons of LIP activity between the CT and IT conditions.

**Figure supplement 2.** There was no systematic relationship between direction preference and saccade-related modulation in LIP neurons' responses to motion stimuli.

**Figure supplement 3.** The comparison of LIP activity between the CT and IT conditions.

**Figure supplement 4.** The motion and saccade representations in LIP shown by dPCA analysis.

**Figure supplement 5.** Modulation of sensory evaluation by saccade selection is unlikely to be an artifact of saccade direction selectivity.

LIP neurons showed significantly elevated activity in the CT relative to the IT at all coherence levels (*Figure 2—figure supplement 1A, B*, nested ANOVA: $P_{(high)}$=0.0326, $F$=4.65; $P_{(medium)}$=0.0088, $F$=7.03; $P_{(low)}$=0.0076, $F$=7.32; $P_{(zero)}$=0.0124, $F$=6.4), and a trend toward lower activity to the nonpreferred direction for CT vs. IT (*Figure 2—figure supplement 1C, D*, nested ANOVA: $P_{(high)}$=0.0994, $F$=2.75; $P_{(medium)}$=0.0649, $F$=3.12; $P_{(low)}$=0.0311, $F$=4.73; $P_{(zero)}$=0.0273, $F$=4.96). Most of the LIP neurons (48 of 83) showed such opposing trends in activity modulation between the preferred and nonpreferred directions (*Figure 2I*). These results indicated a nonlinear modulation of saccade choice on motion DS in LIP, aligned precisely with the response property of each neuron. This is unlikely to be driven by a linear gain modulation of saccade DS. Receiver operating characteristic (ROC) analysis further confirmed significantly greater motion DS in the CT condition than in the IT condition (*Figure 2J, K*; nested ANOVA: $P_{(high)}$=5.0e-4, $F$=12.44; $P_{(medium)}$=9.53e-6, $F$=20.91; $P_{(low)}$=9.33e-7, $F$=26.03; $P_{(zero)}$=2.56e-8, $F$=34.3). Such DS differences were observed even before stimulus onset. Moreover, LIP neurons exhibited similar levels of mean activity between different saccade directions (CT vs. IT) before monkeys' saccade choice (*Figure 2L*), further supporting that saccade DS did not significantly contribute to the observed modulation of LIP neurons' responses to motion stimuli.

To better quantify the influence of saccade direction on neuronal responses to motion stimuli, we calculated a modulation index for each neuron (STAR Methods). We then compared the modulation indices across neurons with different direction preferences and found no systematic relationship between direction preference and saccade-related modulation in LIP neurons at the population level (*Figure 2—figure supplement 2*). Furthermore, a demixed principal component analysis on the pseudo-population activity (STAR Methods) revealed that the saccade direction related representation was a substantial component of LIP activity, as the saccade direction and motion–saccade interaction together explained a similar amount of variance in the population activity as the stimulus motion direction did (*Figure 2M*, *Figure 2—figure supplement 3*). In contrast, we did not observe significant DS differences between the CT and IT conditions in the data sessions in which the saccade targets were aligned close to the vertical direction (21 of 104 neurons, *Figure 2—figure supplement 4*).

We further investigated whether there was a systematic relationship between neurons' RF positions and the influence of saccade direction on motion DS. This was motivated by the intuitive expectation that CTs were more likely to fall within the RFs of LIP neurons than ITs. To examine this, we first quantified the RF centers of all LIP neurons with identifiable RFs during the MGS task (see Materials and methods) and plotted their modulation indices against RF positions (*Figure 2—figure supplement 5A*). We found that neurons with RFs located farther from the horizontal meridian exhibited stronger modulation by saccade direction, whereas those with RFs closer to the meridian showed weaker and more inconsistent modulation. This pattern aligns with the results shown in *Figure 2—figure supplement 3*, which also demonstrated minimal differences between CT and IT conditions when saccade targets were positioned near the vertical axis. Next, for neurons that did not exhibit a clear RF in the MGS task, we aligned their responses based on the saccade direction that evoked the highest mean firing rate and plotted mean responses across all directions (*Figure 2—figure supplement 5B*). These neurons did not show significant differences across directions, confirming the absence of a clear RF structure. Finally, we identified the two saccade targets in the MGS task that were spatially closest to those used in the main task, labeled them as CT and ITs (following the same convention as in the main task), and compared the mean firing rates across all neurons. No significant difference was found between CT and IT conditions (*Figure 2—figure supplement 5C*; mean$_{(ipsi)}$>mean$_{(contra)}$, paired t-test, p=0.053). Together, these results suggest that the nonlinear modulation of sensory evaluation by saccade selection is unlikely to be an artifact of basic saccade DS'.

## Decision-correlated but not stimulus-correlated activity was modulated in LIP

We then examined the impact of nonlinear feedback modulation on the correlation between LIP DS and monkeys' choice behavior. We found that LIP DS was more decision-correlated in the CT condition than in the IT condition. Illustrated in *Figure 2K*, the LIP DS on zero-coherence trials, which reflected monkeys' trial-by-trial categorical choice, was significantly greater in the CT condition than in the IT condition (nested ANOVA: p=2.56e-8, $F$=34.3). We also quantified the correlation between LIP neural activity and the trial-by-trial categorical choice or the physical motion direction by comparing LIP neural activity on correct versus incorrect trials. We used low-coherence trials, as monkeys made

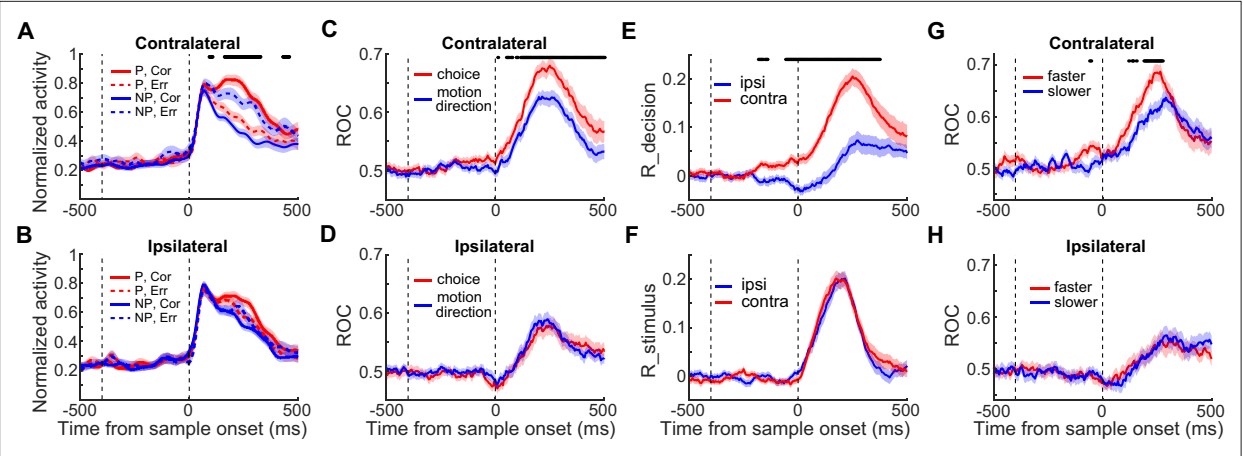

**Figure 3.** Feedback modulation specifically impacted the decision-correlated activity. (**A–B**) Averaged population activities on low-coherence trials in the CT condition (**A**) and IT condition (**B**) are shown separately for correct (corr) and error (err) trials. (**C–D**) An ROC analysis quantified the stimulus-related and decision-related LIP DS on low-coherence trials. The correlations between LIP neural activity and the monkeys' decisions about motion direction (red) or the physical direction of the motion stimulus (blue) in both the CT (**C**) and IT (**D**) conditions are shown over time. (**E–F**) Partial correlation analysis revealed the decision-related and stimulus-related components of LIP activity. The values for r-decision (**E**) (the partial correlation between neuronal activity and monkeys' choice, given the stimulus direction) and r-stimulus (**F**) (partial correlation between neuronal activity and stimulus direction, given the monkeys' choices) were compared between the IT and CT conditions. (**G–H**) Correlation between LIP DS and the monkeys' RTs on zero-coherence trials. The choice selectivity on zero-coherence trials is shown for the faster RT and slower RT trials. Shaded areas denote ± SEM. The black stars indicate time periods in which there was a significant difference (paired t-test: p<0.01).

The online version of this article includes the following figure supplement(s) for figure 3:

**Figure supplement 1.** The comparison of LIP activity to the preferred motion direction between the faster and slower RT trials in the CT condition.

**Figure supplement 2.** The comparisons of LIP activity to the preferred motion direction between the faster and slower RT trials in the IT condition, which are shown in the same format as in *Figure 2—figure supplement 4*.

enough errors in these trials. In the CT condition but not the IT condition, LIP DS at the population level was significantly reversed in sign on incorrect trials as compared to correct trials (*Figure 3A and B*; nested ANOVA: $P_{(CT)}$=0.0045, *F*=8.32). Accordingly, in the CT condition but not in the IT condition, LIP activity correlated more closely with the monkeys' trial-by-trial abstract decisions about motion direction, as opposed to the physical motion direction (*Figure 3C and D*; nested ANOVA, $P_{(CT)}$=0.0016, *F*=10.31; $P_{(IT)}$=0.443, *F*=0.59). Meanwhile, LIP activity correlated more closely with the trial-by-trial abstract decisions during the CT condition relative to the IT condition on the low coherence trials (nested ANOVA: p=1.95e-8, *F*=34.94). Furthermore, we used partial correlation analysis to examine decision- and stimulus-related components of DS (i.e. r-decision and r-stimulus, *Figure 3E and F*) using all four coherence levels. The decision-related component of LIP DS was significantly greater in the CT condition than in the IT condition (*Figure 3E*; nested ANOVA: p=1.07e-6, *F*=25.72), and this difference emerged even before motion stimulus onset. This suggests that the LIP DS was more closely correlated with monkeys' decisions in the CT condition than in the IT condition. The upregulation in r-decision for contralateral choices may reflect the monkeys' internal choice bias or expectation (choice between two motion directions) prior to stimulus presentation, which could influence their subsequent decisions more in the CT condition. However, the stimulus-related component of the LIP DS was similar at the population level between the two conditions (*Figure 3F*; nested ANOVA: p=0.7606, *F*=0.09), suggesting that the modulation of LIP DS by the later saccade choice was primarily related to a decision process rather than basic sensory processing.

We further examined the impact of nonlinear feedback modulation on the correlation between LIP DS and the monkeys' RTs. Shortly after motion stimulus onset, LIP neurons showed greater population response to their preferred motion direction on shorter versus longer RT trials for all motion coherence levels in the CT condition (*Figure 3—figure supplement 1*) but not in the IT condition (*Figure 3—figure supplement 2*). On zero-coherence trials (*Figure 3G and H*), DS evolved more rapidly on shorter vs. longer RT trials in the CT condition but not in the IT condition (bootstrap:

P(CT)<0.01, P(IT)=0.87). Together, these results indicated that the feedback modulation of saccade choice on sensory evaluation predominantly impacted the decision-correlated activity in LIP.

## Trained multi-module RNNs replicated the nonlinear feedback modulation

Our electrophysiology data show that action selection nonlinearly modulates the neural processing of sensory evaluation, which indicates a complex, iterative computation for flexible decision-making. Training RNNs on behavioral tasks used in experimental neurophysiological studies has proven to be helpful for exploring putative circuit computations underlying cognitive tasks (*Zhou et al., 2021*; *Masse et al., 2019*; *Song et al., 2016*). Therefore, we trained multi-module RNNs to perform the FVMD task to examine the circuits and computation mechanisms underlying the interplay of different decision processes. We adopted simple neurobiological principles to constrain the connection structure but not the functional roles to generate recurrently connected modules (STAR Methods). Because our neurophysiological results showed that the modulation effect of action selection on sensory evaluation depended on the saccade direction relative to the recorded brain hemisphere, we implemented RNNs composed of two main modules organized in parallel to simulate two brain hemispheres. Each main module consisted of two nominal modules, with each nominal module receiving the visual motion input (motion module) or the target color input (target module), corresponding to the two neuron populations whose RFs covered the motion stimulus or saccade targets, respectively (*Figure 4A*).

We independently trained 50 such networks with randomly initialized weights and identical hyperparameters to perform the FVMD task. The FVMD task setup used for training RNNs was tailored to match the monkey experiments. Specifically, the RNNs were trained using motion stimuli with two high-coherence levels and were tested using stimuli with another four different coherence levels (high, medium, low, and zero). After training, all 50 networks converged to perform the FVMD task with high accuracies (>99%) on the training coherence and exhibited coherence-dependent performances when tested with untrained stimuli (*Figure 4B and C*): accuracy increased and RT decreased with motion coherence.

We then analyzed the unit activity in different modules when the trained RNNs were tested with untrained motion stimuli. Across all networks, the units in the motion module and target modules exhibited activity corresponding to sensory evaluation and action selection, respectively. Specifically, units in the motion modules showed coherence-correlated motion DS (*Figure 4D, F*, *Figure 4—figure supplements 1 and 2*; mean $r$=0.29; one-sample t-test: p=4.09e-42, t(49) = 46.31), and neuronal activity on zero-coherence motion trials reflected the trial-by-trial abstract decisions of the networks (t-test: P(example RNN)=5.97e-27, t(91) = 15.33; P(population)=1.05e-31, t(49) = 27.86). Such motion DS resembled our electrophysiology data when the motion stimuli were presented within the RFs of the LIP neurons. Meanwhile, units in the target modules showed coherence-correlated saccade DS (*Figure 4E, F*, *Figure 4—figure supplements 1 and 2*), and the neuronal activity on zero-coherence motion trials also showed significant choice probability (t-test: P(example RNN)=7.06e-8, t(68) = 15.83; P(population)=2.08e-22, t(49) = 17.21). Such saccade DS was consistent with the commonly observed decision-related activity in previous studies in which the saccade target was presented within the RFs of LIP neurons (*Roitman and Shadlen, 2002*; *Shadlen and Newsome, 1996*; *Shadlen and Newsome, 2001*).

Furthermore, we found that the motion DS of units in the motion modules was significantly greater in the CT condition than in the IT condition for the majority of RNNs (*Figure 4H and I* and *Figure 4—figure supplement 3A, B*): in the CT condition relative to the IT condition, activity was significantly greater for the preferred motion direction but weaker for the nonpreferred direction (*Figure 4—figure supplement 4*). ROC analysis confirmed that the motion DS in the motion module was significantly greater for all four coherence levels in the CT vs. IT conditions in most RNNs (*Figure 4J, K and N* and *Figure 4—figure supplement 3C, D*; two-way ANOVA: P(example RNN)=1.53e-11, $F$=47.49; P(population)=3.50e-77, $F$=660.84). Similar to the monkey electrophysiology data, the decision-correlated motion DS (r-decision) in the motion module was dramatically reduced in the IT versus CT condition (paired t-test: p=6.84e-17, t(49) = −12.53), and this difference was substantially greater than the difference in the stimulus-related (r-stimulus) motion DS between the two conditions (paired t-test: p=2.02e-14, t(49) = −10.70; *Figure 4L, M and O*, *Figure 4—figure supplement 3F*). These results

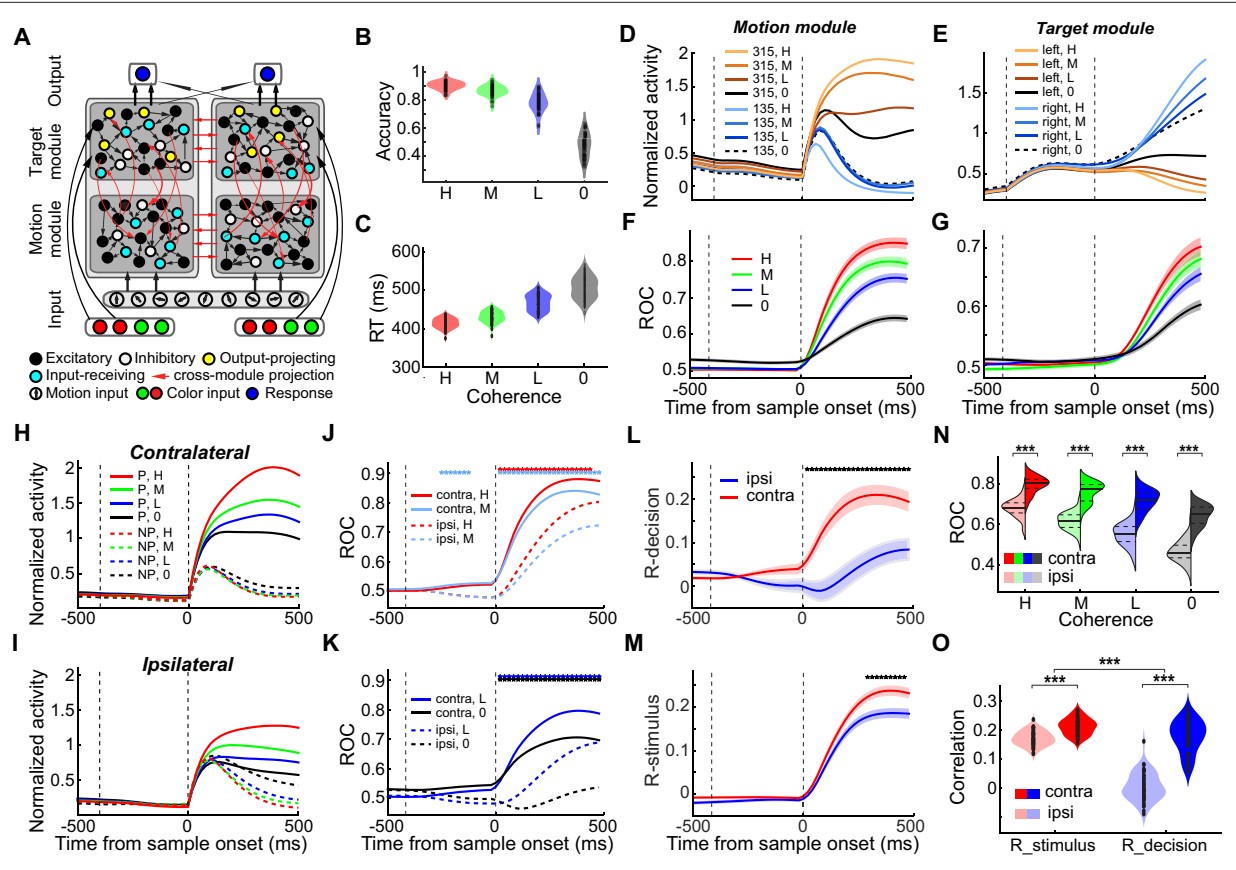

**Figure 4.** Multi-module RNNs trained with the FVMD task. (**A**) Model schematic of the RNNs. Each RNN consists of nine motion direction tuned input units, eight color tuned target input units, 200 hidden units, and two response units. The hidden layer of each RNN consists of two main modules. Each main module consists of two nominal modules, each of which receives either the visual motion input (motion module) or the target color input (target module). Only the target modules project to the two response units, and each main module projects primarily to one response unit. All four nominal modules were assigned with an equal number of units (25%) in the network, which consisted of 80% excitatory and 20% inhibitory units. (**B–C**) The performance accuracies (**B**) and RTs (**C**) of all 50 trained RNNs are shown separately for each motion coherence level. (**D–E**) Two example units from an example RNN. (**D**) The neural activity of an example unit from the motion module is shown for each motion direction and coherence level. (**E**) The neural activity of an example unit from the target module is shown for each saccade direction and coherence level. (**F–G**) The motion DS for the motion module (**F**) and saccade DS for the target module (**G**) for the example RNN. (**H–I**) The averaged population activities of all direction-selective units in the motion module of the example RNN are shown for the CT condition (**H**) and IT condition (**I**). (**J–K**) The averaged motion DS in the motion module of the example RNN for both the CT (solid) and IT (dashed) conditions was quantified by ROC analysis. (**L–M**) Partial correlation analysis. The values for r-decision (**L**) and r-stimulus (**M**) are compared between the IT and CT conditions for the example RNN. (**N**) A comparison of the motion DS in the motion module of all trained RNNs between the CT and IT conditions is shown for each coherence level. (**O**) A comparison of the r-decision and r-stimulus between the IT and CT conditions is shown for all trained RNNs. (Paired t-test: ***, p<0.001).

The online version of this article includes the following figure supplement(s) for figure 4:

**Figure supplement 1.** Examples of unit activity in the example network.

**Figure supplement 2.** Population level unit activity across all trained RNNs.

**Figure supplement 3.** The motion direction selectivity in the motion module was significantly modulated by later saccade choice in the RNNs.

**Figure supplement 4.** The comparison of unit activity in the motion module of the RNNs between the CT and IT conditions.

indicated that action selection also nonlinearly modulated the neural processing of sensory evaluation during decision-making in the RNNs.

## Selectivity-specific feedback connections mediate the modulation of sensory evaluation by action selection

Our multi-module RNNs showed a remarkable similarity to both the behavioral performances and the neural activity patterns in the monkey electrophysiology data, suggesting that these networks may

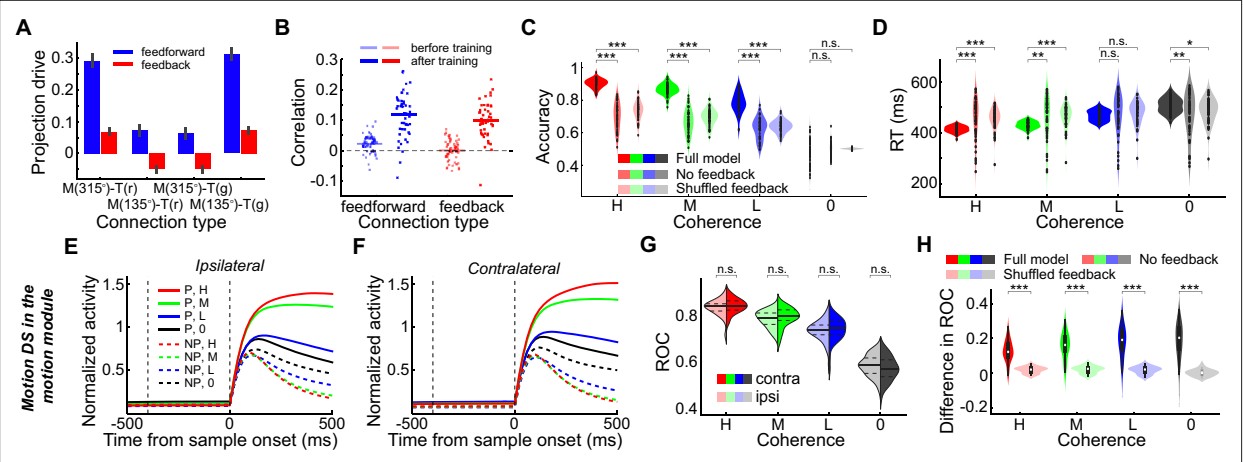

**Figure 5.** The circuit mechanisms underlying the nonlinear feedback modulation in RNNs. (**A**) The averaged cross-module connection weights are shown for both feedforward and feedback connections. Units in the motion module (M) and target module (T) were grouped based on their preferences for motion DS (315° vs. 135°) and target color selectivity (red [r] vs. green [g]), respectively. (**B**) The correlation between the match extent of the neural encoding between units in the motion module and those in target modules and the connection weights between them. Each dot denotes data from one RNN (N=50). (**C–D**) A comparison of the performance accuracy (**C**) and RT (**D**) of the full-model RNNs, RNNs without feedback connections, and RNNs with shuffled feedback connections. (**E–F**) The averaged activity of units in the motion module of the example RNN after the feedback connections were ablated is shown for CT (**E**) IT (**F**) conditions. (**G**) The comparison of the averaged motion DS between CT and IT conditions for all trained RNNs after feedback connections were ablated. An ROC analysis was used to quantify the motion DS. (**H**) The differences in motion DS between CT and IT conditions were compared between full-model RNNs and RNNs with shuffled feedback connectivity.

The online version of this article includes the following figure supplement(s) for figure 5:

**Figure supplement 1.** The motion DS in the motion module of the trained RNNs did not decrease after inactivating feedback connectivity.

**Figure supplement 2.** The averaged population activity of all trained RNNs without feedback connections.

**Figure supplement 3.** The effects of pattern-specific ablation of the feedback connections on the RNNs' behavior performance.

**Figure supplement 4.** The averaged activity of RNNs with disrupted feedback connectivity.

**Figure supplement 5.** The behavioral performance and unit activity in the RNNs that were initialized without feedback connectivity.

**Figure supplement 6.** The performance accuracies of the additional RNNs trained without feedback connectivity are shown separately for each motion coherence level.

have adopted mechanisms similar to those in the monkey brain to mediate the decision processes in the FVMD task. Therefore, we explored the potential circuit mechanisms underlying the modulation of action selection on sensory evaluation in RNNs. We hypothesized that this modulation might result from the feedback connections from the target module to the motion module. To test this hypothesis, we examined the recurrent weights of the cross-module connections (STAR Methods). Both the feedback and feedforward cross-module connection weights were significantly greater between the selectivity-matched unit pairs (e.g. units in the motion module preferred to encode 315° and units in the target module preferred to encode red) than the selectivity-nonmatched unit pairs (e.g. units in the motion module preferred to encode 315° while units in the target module preferred to encode green; *Figure 5A*; paired t-test: $P_{(feedforward)}$=7.18e-16, t(49) = –11.76; $P_{(feedback)}$=5.26e-18, t(49) = –13.40). In particular, the average feedback connectivity was primarily excitatory between the selectivity-matched unit pairs but was inhibitory on average between the selectivity-nonmatched unit pairs. Furthermore, most of the networks exhibited positive correlations between the connection weight and the match extent of the neural selectivity of each cross-module unit pair (*Figure 5B*; mean *r*=0.11; one-sample t-test: p=6.39e-16; t(49) = 11.79; STAR Methods), indicating that units that exhibited stronger encoding preference to one motion direction in the motion module were more likely to receive more extensive feedback projections from the units that exhibited stronger encoding on the matched target color in the target module. These results suggested precise feedback connections between RNN modules that were aligned with the functional properties of different units.

To test the causal contribution of the selectivity-specific feedback connections to the modulation of action selection on sensory evaluation, we conducted three projection-specific inactivation

experiments in the RNNs (STAR Methods). First, we performed nonspecific ablation of all the feedback projections from the target module to the motion module when testing with untrained motion stimuli. In most of the tested RNNs, this caused a dramatic reduction in their performance in all nonzero-coherence stimulus conditions (*Figure 5C*; two-way ANOVA: p=4.74e-80, *F*=704.65) as well as significantly prolonged RTs for high- and medium-coherence stimulus conditions (*Figure 5D*; two-way ANOVA: p=2.45e-4, *F*=13.79). The mean RTs became more dispersed across all the trained RNNs and did not differ among different motion coherences after ablation (one-way ANOVA: p=0.68, *F*=0.50). Ablating feedback connections significantly affected the motion DS in the motion module. On average, although the averaged motion DS did not decrease after ablation (*Figure 5—figure supplement 1A, C*), the difference in the motion DS between the CT and IT conditions vanished (*Figure 5E–G* and *Figure 5—figure supplement 2A, B*; two-way ANOVA: $P_{(CT\ vs.\ IT)}$=0.14, *F*=2.21). These results indicated a dramatic reduction in the nonlinear feedback modulation on sensory evaluation by action selection in the RNNs. To further test the importance of different patterns of feedback connectivity for the RNNs to solve the FVDM task, we next performed pattern-specific ablation of the feedback connections in RNNs. We separated the feedback projections in each RNN into specific (i.e. the feedback connection weights were positive or negative between the selectivity-matched or selectivity-nonmatched unit pairs, respectively) and nonspecific groups. Despite similar numbers and weights of the feedback connections in these two groups (*Figure 5—figure supplement 3A, B*), ablating the feedback connection in the specific group versus the nonspecific group resulted in much more severe impairments in the RNNs' behavior performance (*Figure 5—figure supplement 3C, D*). These results indicated that the feedback connections that reflected the learned stimulus–response association were crucial for the RNNs to solve the FVDM task.

Second, we disrupted the selectivity-specific feedback connectivity without changing the total strength of the feedback connections by randomly rearranging the feedback connection weights. This also resulted in a significant reduction in the RNNs' performance accuracy for nonzero-coherence motion stimuli (*Figure 5C*; two-way ANOVA: p=1.38e-90, *F*=883.61) and prolonged RTs for high- and medium-coherence motion stimuli (*Figure 5D*; two-way ANOVA: p=1.94e-12, *F*=54.06). The changes of RNN units' activity patterns after scrambling feedback connections were similar to the effects after feedback connectivity was fully ablated (*Figure 5—figure supplements 1B, C and 4A, B, D and E*). Particularly, the levels of nonlinear modulation of sensory evaluation by saccade selection dramatically decreased, as evident by the diminished difference in the motion DS between the CT and IT conditions (*Figure 5H*, two-way ANOVA: p=5.81e-12, *F*=51.52).

Third, we trained another 50 RNNs without feedback connectivity to learn the FVMD task. Similar to the full-model RNNs, all 50 networks converged to perform the FVMD task with high accuracies (>99%) after training and exhibited coherence-dependent performances when tested with untrained stimuli (*Figure 5—figure supplement 5A, B*). Meanwhile, units in the motion module and target modules exhibited coherence-dependent motion DS and saccade DS, respectively (*Figure 5—figure supplement 5C, D*). Consistent with the above connectivity ablation experiment, units in the motion module of these RNNs did not exhibit different levels of motion DS between CT and IT conditions (*Figure 5—figure supplement 5E–G*; two-way ANOVA: p=0.63, *F*=0.24), indicating no noticeable nonlinear feedback modulation on sensory evaluation by action selection. Importantly, the performance accuracies of these RNNs decreased significantly when tested with untrained nonzero-coherence stimuli (*Figure 5—figure supplement 5H*; two-way ANOVA: p=1.46e-14, *F*=65.64), as compared to the full-model RNNs. Furthermore, we trained an additional 50 RNNs without feedback connections to learn the FVMD task. These RNNs were initialized with greater recurrent connection probabilities than the full-model RNNs, such that the number of total trainable connection weights matched that in the full-model RNNs. Interestingly, these RNNs still exhibited significantly lower performance accuracies when tested with high- and medium-coherence stimuli relative to the full-model RNNs (*Figure 5—figure supplement 6*; two-way ANOVA: p=8.24e-6, *F*=20.60). Together, these results suggested that the nonlinear feedback modulation, which was mediated by the selectivity-specific feedback connections, was important for the RNNs to efficiently solve flexible decision tasks.

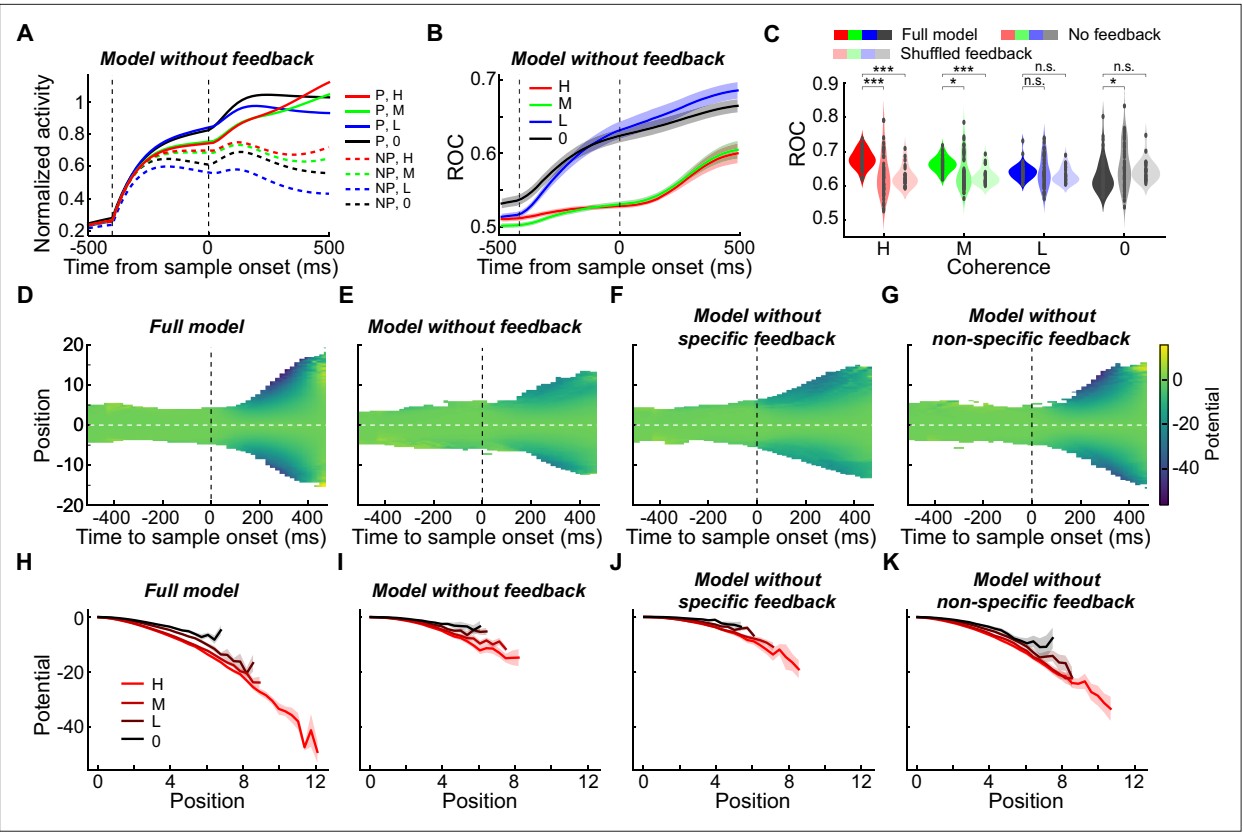

**Figure 6.** The computational mechanisms underlying the nonlinear feedback modulation in decision-making. (**A**) The averaged activity of units in the target module of the example RNN, after the feedback connections were ablated, is shown separately for different saccade directions and coherence levels. (**B**) The averaged saccade DS in the target module of the example RNN after full feedback ablation. (**C**) The averaged saccade DS in the target module of the full-model RNNs, RNNs without feedback connections, and RNNs with shuffled feedback connections. (Paired t-test: *, p<0.05; **, p<0.01; ***, p<0.001; n.s., not significant). (**D–G**) The evolution of the averaged energy landscapes was shown over time. A numerical approximation of the energy landscape in the 1-D decision (saccade choice) subspace is constructed for both full-model RNNs and RNNs with various types of projection-specific inactivation. Each plot represents the averaged results of 50 RNNs, where Position 0 signifies the SVM decision boundary, and the vertical dashed line marks the time of motion stimulus onset. Unvisited portions of the state space are left blank as there is no gradient or potential estimate. (**H–K**) Averaged numerical estimate of energy landscapes for trials with different task difficulty levels (motion coherence) (N=50). Results from two different saccade choices were averaged together. Only the potential values at positions continuously visited by four or more models were retained. Shaded areas denote ± SEM.

## Nonlinear feedback modulation enhanced the decision consistency by strengthening the attractor basins of network dynamics

Our RNN modeling indicates a crucial role for nonlinear feedback modulation in optimizing flexible decision-making. Subsequently, we delved into the computational mechanisms embodied by nonlinear feedback modulation during the decision-making process. Specifically, we examined the unit activity in the target module, a direct contributor to the decision outputs of RNNs. Strikingly, the projection-specific inactivation experiments led to a significant reduction in the magnitude of saccade DS in the nonzero-coherence trials (*Figure 6A–C*, $P_{(ablating\ feedback)}$=6.90e-5, $F$=16.29; $P_{(shuffling\ feedback)}$=1.2e-19, $F$=95.08; two-way ANOVA). Additionally, saccade DS emerged earlier than the motion stimulus onset in the nonzero-coherence trials, and its positive correlation with the motion coherence levels diminished after the projection-specific inactivation (*Figure 6A–C* and *Figure 5—figure supplements 2C and 4C, F*; mean $r_{(ablating\ feedback)}$=–0.034; mean $r_{(shuffling\ feedback)}$ = –0.030). These disrupted patterns of saccade DS observed in the target module following projection-specific inactivation aligned with the decreased decision consistency of RNNs, where decision consistency reflects the degree of agreement in the model's choices under specific task conditions. This suggests a diminished reliance on sensory input and an increased dependence on internal noise in the decision-making process.

Previous studies have demonstrated that attractor dynamics in networks can explain choice consistency, with steeper landscapes around attractor basins reflecting consistent decisions (*Deco et al., 2013*; *Finkelstein et al., 2021*; *Wang et al., 2023*; *Wang, 2002*; *Wong et al., 2007*). Consequently, we investigated attractor dynamics in the state space of network activity that underlies decision-making variability in RNNs with different types of feedback connectivity. Similar to prior studies (*Wang et al., 2023*), we examined the neural dynamics underlying decisions in a 1-D neural subspace using unit activity in the target module responsible for saccade choices (see Materials and methods). Beginning with the reconstruction of a numerical approximation of the energy landscape in this 1-D subspace, we explored changes in this landscape after different types of projection-specific inactivation. This revealed a dramatic reduction in the depth of attractor basins in the energy landscape of population activity after ablating all feedback connectivity (*Figure 6D–E*, paired t-test: p=2.28e-11, t(49) = –8.61). Moreover, ablating the feedback connections led to a more severe reduction in the depth of attractor basins in the network dynamics within the specific group compared to the nonspecific groups (*Figure 6F–G*, paired t-test: p=1.59e-6, t(49) = -5.46).

We further examined how the energy landscape in the 1-D subspace changed in relation to task difficulty (motion coherence). Consistent with prior findings (*Wang et al., 2023*), trials with lower decision consistency (trials using lower motion coherence) exhibited shallower attractor basins at the time of decision for all types of RNNs (*Figure 6H–K*). However, both the depth and the positional separation of attractor basins in the network dynamics significantly decreased for all non-zero motion coherence levels after the ablation of all feedback connections (comparing *Figure 6I* with *Figure 6H*; $P_{(depth)}$=5.20e-25, $F$=122.80; $P_{(position)}$=1.82e-27, $F$=137.75; two-way ANOVA). Notably, this reduction in basin depth and separation was more pronounced in the specific group compared to the nonspecific groups after ablating the feedback connections (comparing *Figure 6J* with *Figure 6K*; $P_{(depth)}$=2.65e-13, $F$=57.35; $P_{(position)}$=3.73e-14, $F$=61.79; two-way ANOVA). These results might underlie the computational mechanisms that explain the observed reduction in the decision consistency of RNNs following projection-specific inactivation: the shallower and closer attractor basins after ablating feedback connections resulted in less consistent decisions. This happened because the variability in neural activity made it more likely for population activity to stochastically shift out of the shallower basins and into nearby alternative ones.

## Discussion

Here, we show that primate LIP activity related to evaluating the visual motion stimulus was nonlinearly modulated by an impending saccade choice that was used to report decisions during a FVMD task, even though the saccades were toward non-RF locations. This suggests that the sensory evaluation and action selection may proceed iteratively during decision-making. This view is also consistent with the common observation that action selection-related activity arises at the early stage of evidence accumulation during decision-making (*Gold and Shadlen, 2007*; *Roitman and Shadlen, 2002*; *Shushruth et al., 2018*), as well as our observation that the modulation of action selection on sensory evaluation emerged during the early task period even before motion stimulus onset in the FVMD task. Brain areas that are related to different decision processes, although spatially separated (*Zhou and Freedman, 2019*; *Katz et al., 2016*; *Jun et al., 2021*), likely form a real-time associated network to solve current decision tasks. The instantaneous result of sensory evaluation may be transmitted to the brain areas responsible for action selection during decision-making in a feedforward manner, and the action selection process may also exert a real-time modulation on the neural processing of sensory evaluation in a feedback manner.

The feedback modulation we observed during decision-making was distinct from the modulation of feature-based attention in early sensory cortex. Although both modulation effects match the stimulus tuning of recipient cells (*Motter, 1994a*; *Motter, 1994b*; *Martinez-Trujillo and Treue, 2004*; *Bichot et al., 2005*; *Maunsell and Treue, 2006*), only the feedback modulation observed in the current study reflects the neural processing of decision-making. This is because only the decision-related component, not the stimulus-related component, of LIP activity was modulated by the monkeys' later saccade choice during the FVMD task (*Figure 3*). This suggests that the feedback modulation from action selection to LIP activity was primarily related to the decision process rather than low-level sensory processing.

A recent study demonstrated that neurons in the middle temporal area responded more strongly to motion stimuli when monkeys saccaded toward their RFs in a standard decision task with a fixed mapping between motion stimuli and saccade directions (*Laamerad et al., 2024*). This modulation emerged through the training process and contributed causally to the monkeys' following saccade choices. Consistently, we found that the response of LIP neurons to motion stimuli was more strongly correlated with the monkeys' decisions in the CT condition (saccades toward RFs) than in the IT condition, in a more flexible decision task. Together, these results suggest that the modulation of action selection on sensory processing may be a general process in perceptual decision-making. However, the observed modulation of saccade direction on LIP neurons' responses to motion stimuli cannot be simply explained by saccade DS. Several lines of evidence argue against this possibility. First, the modulation effect was nonlinear; specifically, neuronal firing rates increased for preferred motion directions but decreased for non-preferred directions (*Figure 2I*, *Figure 2—figure supplement 1*). This pattern is unlikely to be driven by a linear gain modulation based on saccade directions. Second, we found that LIP neurons exhibited similar levels of activity in both the CT and IT conditions (*Figure 2M*), which is inconsistent with the presence of clear saccade DS.

The selectivity-specific feedback modulation from action selection to sensory evaluation emerged in our multi-module RNNs during training, although the cross-module connections were randomly initialized before training. Importantly, our causal in-silico experiments demonstrated that these precise feedback connections played an essential role in mediating the modulation of action selection on sensory evaluation while solving the FVDM task. Previous studies showed that both the corticocortical feedback from the secondary visual cortex to the primary visual cortex and the corticogeniculate feedback in primates are organized into parallel streams resembling the reciprocal feedforward pathways (*Federer et al., 2021*; *Briggs and Usrey, 2009*; *Briggs, 2020*), suggesting a potential functionally specific feedback connection in visual processing. However, to the best of our knowledge, a similar segregation of feedback that reflects the encoding properties of the target neurons has not been evident in the decision network in vivo. Therefore, it will be important for future studies to examine the connectivity and correlation of neural activity among neural groups related to different decision processes through sophisticated anatomical experimental approaches as well as multi-channel recordings in vivo.

Our brain includes extensive feedback connections across different brain areas, which, in some cases, even outnumber the feedforward connections (*Briggs, 2020*; *Harris and Mrsic-Flogel, 2013*). However, compared to the feedforward connections, much less is known about the function of feedback connections. Feedback modulation has been implicated in top-down modulation of neuronal responses in the early sensory cortex, such as attention and expectation, which facilitate the processing of important stimuli and suppress distractors (*Moore and Armstrong, 2003*; *Gilbert and Li, 2013*; *McManus et al., 2011*). Furthermore, both experimental and computational studies have shown that weak decision-correlated neural activity (i.e. choice probability) as well as the correlated firing of pairs of neurons (i.e. noise correlation) in the early sensory cortex might partially result from feedback input in decision tasks (*Nienborg and Cumming, 2009*; *Bondy et al., 2018*; *Wimmer et al., 2015*), suggesting a potential role for feedback connection in modulating early sensory processing during decision-making. Consistently, the nonlinear feedback modulation, mediated primarily by selectivity-specific feedback connections, was important for our multi-module RNNs to efficiently solve the FVMD task, as the RNNs' decision behavior became more stochastic following ablating/disrupting the feedback connectivity. This was evident not only in the diminished behavioral performance of the RNNs but also in the disrupted patterns of activity related to saccade choice in the target module. Notably, the nonlinear feedback modulation intensified the attractor basins of the population activity associated with saccade choice, led to more reliable decisions based on sensory input. Our results unveil a novel pattern of feedback modulation during the neural processing of decision-making and suggest a potentially critical role that feedback modulation plays in increasing the consistency of flexible sensorimotor decisions.

Two-state attractor models have been widely employed to elucidate two-alternative forced decision-making processes (*Deco et al., 2013*; *Finkelstein et al., 2021*; *Wang et al., 2023*; *Wang, 2002*; *Wong et al., 2007*; *Prat-Ortega et al., 2021*; *Wong and Wang, 2006*). In these models, decisions arise when network activity falls into one of two attractor basins. Previous studies have posited that decision consistency is influenced by the configuration of the energy landscape surrounding these

attractor basins (*Wang et al., 2023*; *Wong et al., 2007*; *Wong and Wang, 2006*). Essentially, steep-sided and deep-basin energy landscapes contribute to consistent decision-making, as it becomes challenging for internal noise to shift activity between basins. Conversely, when energy landscapes are relatively flat and attractor basins are shallow, decisions are generated more stochastically, as it is easier for internal noise to drive activity out of the attractor basins and into alternative ones. Consistently, our RNNs modeling showed that shallower and less separated attractor basins were associated with decreased decision consistency in more difficult trials. Building upon previous research, our results provide in silico evidence supporting the notion that the energy landscape within the state space of population activity causally influences decision consistency. Moreover, our findings present the initial circuit mechanism through which the configuration of the energy landscape for decision-related activity is modulated by feedback modulation, to our knowledge. Future studies are needed to explore the connections between different types of circuit connectivity and population activity dynamics in cognitive processing in vivo.

Our results are consistent with predictive coding theories of feedback function, which propose that an internal model of the world is generated in the brain based on sensory data and prior experience and is refined by incoming sensory data (*Rao and Ballard, 1999*; *Friston and Kiebel, 2009*; *Bastos et al., 2012*; *Shipp, 2016*). In terms of the architecture of hierarchical predictive coding schemes, different neuronal ensembles encode different attributes/choices in either the action selection or sensory evaluation process at each level of the cortical hierarchy during flexible decision-making. Meanwhile, the conditionally independent expectations are functionally segregated, so that descending predictions from the action selection process are limited to the sensory evaluation process on the matched/associated stimulus attribute but not the opposite attribute. The selectivity-specific feedback connectivity in our RNNs fits the functional segregation of expectation very well during the interplay between distinct neural processes of decision-making. Specifically, the neural ensembles related to either sensory evaluation or action selection processes for the same choice might form precise recurrently connected loops. The prediction signal carried by the precise feedback connections might facilitate sensory evaluation of the associated stimulus but prohibit evaluation of the nonmatched stimulus. This could, in turn, amplify the task-relevant sensory input, facilitate the sensorimotor transformation, and ultimately result in faster and more accurate action choice in flexible decisions. In future work, it will be important to use techniques such as projection-specific inactivation and microstimulation in vivo to test the causal contribution of feedback connectivity in decision-making and other cognitive functions.

## Materials and methods
### Behavioral task, stimulus display, and animal preparation
The flexible visual motion discrimination (FVMD) task (*Figure 1A*) has been reported previously (*Zhou and Freedman, 2019*) and is briefly summarized below. In this task, monkeys were required to saccade to either the green or red targets based on the direction of the visual motion stimulus. Two motion directions (135°, 315°) were used, each with four different coherence levels (0%, 9%, 18%, 36% for monkey M; and 0%, 13%, 25%, 50% for monkey B) were tested. If the sample direction was 135°, the monkeys must saccade to the green target to receive a juice reward, whereas the 315° direction was associated with the red target. The rewarded target (red or green) was randomly chosen (with 50% probability) on each zero-coherence trial. The positions of red and green targets were randomly chosen between the two positions on each trial at each recording session. Therefore, there is no fixed mapping between motion stimulus and saccade direction.

To initiate each trial, monkeys must hold a touch bar and acquire gaze fixation. They then need to maintain fixation within a 2.0–2.5° window throughout the trial before their saccade choice. After a 500ms fixation period, two colored saccade targets appeared simultaneously at opposite positions relative to the fixation point with equal eccentricities (8° and 9° for Monkey M and B, respectively). 400ms later, a sample motion stimulus was presented at a location orthogonal to the axis of, but at the same eccentricity as, the saccade targets. We used motion stimuli that were full contrast, 8° diameter, random-dot movies composed of 190 dots per frame, and the dots moved at 10°/s. Monkeys needed to saccade to either red or green targets within a 60–2000ms window after sample stimulus onset. The two saccade targets were equidistant from the stimulus, with the distance typically ranging from 12 to 15 degrees. A juice reward would be delivered to the monkeys if they made correct saccade choice.

Two male monkeys (*Macaca mulatta*, 15–16 years old, 8–14 kg) were trained on the FVMD task and implanted with a head post as well as a recording chamber positioned over PPC. Our surgical, behavioral, and neurophysiological approach has been described in detail in a previous study (*Zhou and Freedman, 2019*). Stereotaxic coordinates for chamber placement were determined from magnetic resonance imaging (MRI) scans obtained before chamber implantation. LIP chambers were centered on the intraparietal sulcus (IPS), 4.0 mm posterior to the intra-aural line and 1.0 mm lateral from the midline for monkey M, and 0 mm anterior to the intra-aural line and 15.0 mm lateral from the midline for monkey B. Monkeys were housed in individual cages under a 12 hr light/dark cycle. Behavioral training and experimental recordings were conducted during the light portion of the cycle. Monkeys sat comfortably while head-fixed in a custom-made primate chair inside a dark experiment rig. The task stimuli were displayed on a 21-inch color CRT monitor (1280*1024 resolution, 75 Hz refresh rate, 57 cm viewing distance). Both monkeys were tested with identical stimuli and timing. A solenoid-operated reward system was used to deliver juice reward to the monkeys. Monkeys' eye positions were monitored by an optical eye tracker (SR Research) at a sampling rate of 1 kHz and stored for offline analysis. Stimulus presentation, task events, rewards, and behavioral data acquisition were accomplished using an Intel-based PC equipped with MonkeyLogic software running in MATLAB (http://www.monkeylogic.net). All experimental and surgical procedures were performed in strict accordance with the recommendations in the Guide for the Care and Use of Laboratory Animals of the National Institutes of Health. All of the animals were handled according to approved institutional animal care and use committee (IACUC) protocol #71887 of The University of Chicago.

## Electrophysiological recording

We used either 75 µm tungsten microelectrodes (FHC, ~1 MΩ) or 16-channel V-Probes (Plexon) to record single neuron activity in LIP. Neurophysiological signals were amplified, digitized, and stored for offline spike sorting (Plexon) to verify the quality and stability of neuronal isolations. We recorded neuronal activity in an MGS task to map LIP RF locations before each FVMD recording session.

Our LIP recordings targeted different hemispheres in the two monkeys (monkey M: left hemisphere; monkey Q: right hemisphere). Therefore, the CT condition and IT conditions in *Figure 1F and G* referred to opposite saccade directions (target locations) between the two monkeys. For monkey M, the CT condition corresponded to the trials in which the correct saccade target was on the right visual field, whereas the CT condition for monkey Q referred to the trials in which the correct saccade was toward the left visual field. We localized LIP in each monkey according to the patterns of neuronal activity in the MGS task (i.e. spatial selectivity during stimulus presentation and delay). All neurons included in the dataset were recorded from the same locations (the same grid holes and similar depths: ~5–10 mm from the cortical surface) where we encountered spatial selectivity in the MGS task. LIP neurons were also identified based on anatomical criteria, such as the location of each electrode track relative to that expected from the MRI scans, the pattern of gray–white matter transitions encountered on each electrode penetration, and the relative depths of each neuron.

We aimed to present the visual motion stimulus inside the RFs of the identified neurons during each recording session. For single-channel electrode recording, only neurons that exhibited visual responses to the motion stimuli during prescreening (tested with the FVMD task) were recorded with sufficient trials (~300–600) of the FVDM task. For neurons exhibiting clear spatial RFs during the MGS task, we presented the motion stimulus inside LIP neurons' RFs, while for those neurons that did not show a clear RF during the MGS task, we presented motion stimuli in the positions (always in the visual field contralateral to the recorded hemisphere) in which neurons exhibited the strongest response to the motion stimuli. For the multi-channel recordings, we recorded all neurons isolated across all channels and presented the motion stimulus inside one of the isolated neurons' RFs. Because adjacent recording sites were located 100 µm apart, nearby neurons typically showed similar RF locations.

## Data analysis

### Neuronal pre-screening

While part of the neural data was presented in a previous report (*Zhou and Freedman, 2019*), the current analysis focuses on a different phenomenon which was not examined previously. We included all neurons recorded from single-channel electrodes for the analysis. For multi-channel recordings, we only included the neurons which showed significant modulation (different from fixation period activity,

one-way ANOVA, p<0.01) of their averaged activity across all motion stimuli because the stimulus could not always be presented within the RF of all the recorded neurons. The low-firing neurons whose maximum firing rates were less than 2.0 spikes/s (to the direction producing greater average responses) during stimulus presentation were also excluded for further analysis. We then applied a one-way ANOVA test to compare activity between the two different motion directions during the period following motion stimulus onset (50–250ms after motion stimulus onset) to select neurons that showed significant motion DS during the decision period. Only neurons that showed significant (p<0.01) DS were used for further analysis.

## Determination of RF

To determine each neuron's RF, we analyzed the average firing rates during both the target presentation and delay periods of the MGS task. The RF centers of neurons with significant RFs were determined through a two-step process. First, we selected neurons that exhibited significant RFs in the MGS based on the following criteria: (1) there must be a significant activity difference across the eight target locations and (2) the mean activity during the selected periods should be significantly greater than the baseline activity during the fixation period. Second, for neurons meeting these criteria, we fitted their responses across the eight conditions to a Gaussian function, using the center of the fitted distribution as the RF center.

## ROC analysis

To quantify each neuron's DS in the FVMD task, we applied an ROC analysis to the distribution of firing rates within sliding windows (100ms width, 5ms steps). The ROC value, which ranges between 0.0 and 1.0, indicates the performance of an ideal observer in assigning motion direction based on each neuron's trial-by-trial firing rates. Values of 1.0 and 0.0 correspond to perfect classification (i.e. strong DS), while a value of 0.5 indicates chance-level classification performance (i.e. no DS). For trials with zero coherence motion, we assigned direction labels on each trial based on the monkey's choice.

In *Figure 3*, in order to test whether DS reflected monkeys' trial-by-trial choices, we used an ROC analysis to quantify whether LIP activity correlated with monkeys' trial-by-trial categorical choices more than the physical direction of motion stimulus by analyzing both correct and error trials. We only included low coherence trials in this analysis because there were sufficient numbers of error trials (average performance: Monkey M: 74% correct, Monkey B: 67% correct). LIP neuronal activity was analyzed by ROC according to either the monkey's trial-by-trial categorical choices or the direction of the sample stimulus on each trial. Furthermore, for this analysis, we only used neurons for which we recorded sufficient trials (for both correct and error trials, N>4) for the low coherence condition of each motion direction (n=45).

## Partial correlation analysis

We performed a partial correlation analysis to quantify the correlation between LIP neural activity and the monkeys' trial-by-trial categorical choices or the physical motion direction of the stimulus using all trials. For each trial, we obtained three parameters for the calculation: the stimulus direction, the pre-choice neuronal activity, and the monkeys' choice. We assigned the stimulus directions with different values for different directions and coherence levels: positive and negative values were used for preferred and nonpreferred directions, respectively, while 4, 2, 1, and 0 are used for coding the high, medium, low, and zero coherence levels. We also assigned different values to different choice directions (−2 for choosing the preferred direction, and +2 for choosing the nonpreferred direction). Two measures were then calculated: r stimulus = r (neuronal activity, stimulus direction| choice direction), the partial correlation between neuronal activity and stimulus direction given the monkeys' choices; and r choice = r (neuronal activity, choice direction | stimulus direction), the partial correlation between neuronal activity and the monkeys' categorical choice given the stimulus direction. In *Figure 3*, we used the mean activity within a sliding window (100ms width, sliding with 5ms steps) for each neuron to perform the partial correlation analysis.

## Modulation index

We have calculated a modulation index for each neuron to reflect the influence of saccade direction on neuron's response to visual stimuli. The modulation index is calculated as:

$$\delta_{contra} = r_{pref}^{contra} - r_{non-pref}^{contra}$$

$$\delta_{ipsi} = r_{pref}^{ipsi} - r_{non-pref}^{ipsi}$$

$$\text{MI} = \frac{\delta_{contra} - \delta_{ipsi}}{\delta_{contra} + \delta_{ipsi}},$$

where $r_{pref}^{contra}$ represents the average firing rate from 50ms to 250ms after sample onset for all contralateral saccade trials with a neuron's preferred moving direction of visual stimuli. The naming conventions are the same for $r_{non-pref}^{contra}$, $r_{pref}^{ipsi}$, and $r_{non-pref}^{ipsi}$. An MI value between 0 and 1 indicates higher modulation in contralateral saccade trials, and an MI value between −1 and 0 indicates higher modulation in ipsilateral saccade trials.

## dPCA analysis

We conducted demixed principal component analysis using the methodology and the code from a previous study (*Kobak et al., 2016*; *Brendel et al., 2020*) to reduce the dimensionality of the population activity as the standard PCA and demixes all task variables. Specifically, we tested how much each task variable (motion direction of sample stimuli, saccade direction, motion-saccade interaction, timing) contributes to the LIP population activity during the DMC task.

As demonstrated in the previous study, the dPCA finds separate decoder (F) and encoder (D) matrices for each task variable (∅) by minimizing the loss function:

$$L_{dPCA} = \sum_0 \|X_0 - F_0 D_0 X\|^2$$

where X is a linear decomposition of the data matrix, which contains the instantaneous firing rate of the recorded neurons, into variable-specific averages:

$$X = \sum_0 X_0 + X_{noise}$$

Here, we decomposed the neural activities into four parts: condition-independent, stimulus-dependent (two motion directions with four coherence levels), saccade-dependent (two saccade directions), dependent on the stimulus-saccade interaction, and noise. The decoder and encoder axes permit us to reduce the data into a few components capturing the majority of the variance of the data dependent on each task variable.

## Recurrent neural network (RNN) training

### Network implementation

We trained biologically inspired networks to perform the FVMD task using methodology similar to previous studies (*Zhou et al., 2021*; *Masse et al., 2019*). We implemented multiple modules in the networks through constraints on the input/output structure and the initial recurrent connectivity of the hidden layer. Specifically, we built excitatory-inhibitory networks with 200 hidden units divided into two equal-size main modules to simulate two brain hemispheres. Each main module was further divided into two nominal modules, with each one only receiving the visual motion input (motion module) or target color input (target module), respectively. This design was intended to simulate the two neuron populations whose RFs covered the motion stimulus or saccade target in the monkey electrophysiology experiment (*Figure 4A*). Every nominal module was allocated one quarter of the excitatory units (40) and one quarter of the inhibitory units (10) in the overall network to ensure that the modules did not differ in their balance of excitation/inhibition prior to training. Meanwhile, in order to simulate the local and long-range connection structures in the brain, we set different levels of recurrent connectivity within and between different modules: the local (connectivity within each nominal module) recurrent connection density (probability) was the highest (50%); the across-RF (connectivity between the two nominal modules within the same main module) connection density was set to be

in the medium level (25%); the cross-hemisphere (connectivity between either the motion modules or the target modules across different main modules) connection density was the lowest (10%). Only excitatory neurons could have 'cross-hemisphere' projections to the corresponding nominal module (e.g. from the motion module of the 'left hemisphere' to the motion module of the 'right hemisphere'). Only the connection weight within the hidden layer was updated during training following methods described previously (*Zhou et al., 2021*). In addition, the connection weights between network units are endowed with short-term synaptic plasticity, which is aimed to provide connection weights with activity-dependent fluctuation over short timescales within each trial.

The input to the network consisted of two parts: 9 motion input units and 8 target color input units. The preferred directions of motion input units were evenly spaced across 360°, with response tuning distributed according to a von Mises function. The value of the nth motion input unit was set to

$$r_n = \frac{2}{3}\left(\mathrm{a}ce^{\left(\kappa\cos(\theta-\theta_{pref})\right)} + b\mathcal{N}\left(0,2\right) + \sqrt{\frac{2}{\alpha}}\mathcal{N}\left(0,\sigma_{in}\right)\right) + \mathrm{d},$$

where $\alpha = \frac{\Delta t}{\tau_{\mathrm{mem}}}$ , $\theta$ is the direction of motion stimulus in radian, $\theta_{pref}$ is the preferred direction of this motion input unit, $\mathrm{a} = \frac{4}{e^\kappa}$ after stimulus onset but 0 elsewhere, $c \in [0,1]$ is the coherence of the dot motion stimulus, $b$ is a binary value that equals 0 before stimulus onset and 1 after that, and $d$ is a constant visual input signal with an amplitude of 2 that decays with time. Specifically, for zero-coherence input, since $ace^{\left(\kappa cos(\theta-\theta_{pref})\right)}$ would be zero, we set this term as $0.4max\left(c\right)$.

The target input units were initially evenly divided into two groups projected to the two target modules of hidden units. Each group was further divided into red and green subgroups. The color subgroup was set as active with an amplitude of 4/3 to represent the color within the projected target module, while the other color group remained silent (amplitude =0). An exponential decay filter was also used to fit the activity of sensory neurons in the early sensory cortex across time:

$$\mathrm{f}\left(x\right) = \left(\frac{1}{3}\right)^t \mathrm{x},$$

where $x$ is the constant visual input for motion input and the color signal for color input, $t$ is time in miliseconds.

The two output units of the network simulate two different saccade directions. In order to simulate the oculomotor control in the brain, each brain hemisphere (main module) projects much more densely to the contralateral response units (probability = 0.32) than the ipsilateral response units (probability = 0.08). To generate a probability distribution over output values at each time point, we also applied a softmax function to scale the response unit activities in the output layer.

To reduce the stochasticity of the network activity brought by input and output connections, we randomly sampled input and output weights from a normal distribution ($\mathfrak{N}\left(0.2, 0.05\right)$). In addition, the input weights were re-initialized if there were extreme values (the minimum value was smaller than ¾ of the maximum value) after multiplying the stimulus signal and weight of motion or target input. Similarly, the output weights were re-initialized if there were extreme values in the output weight values. The input units had excitatory and random projections to the recurrent units with a probability of 0.32. However, only the excitatory units in the target modules of each "hemisphere" could project to the output units. Both the input and output weights were fixed for each network during task training.

## Network training

The networks were trained using BrainPy *Wang et al., 2022* on an Intel(R) Core(TM) i9-9900K CPU (3.60 GHz, 8 cores). The network was optimized using backpropagation through time (BPTT) and stochastic gradient descent with an Adam optimizer (default setting, first moment estimates decay rate = 0.9, second moment estimates decay rate = 0.999) to minimize a loss function. The network parameters (recurrent weights/biases) were optimized to minimize a loss function with three parts as in a previous study *Zhou et al., 2021*: (a) a performance loss; (b) a metabolic cost on mean neuronal activity; and (c) a metabolic cost on connectivity. The gradient was clipped to a maximum L2 value of 0.1 to avoid the exploding gradient problem. During training, only the hidden unit-related parameters (weights, bias, and initial activities) were updated. The training process was terminated when the network's performance accuracy reached 99% or until the maximum number of iterations (2000).

During each trial of the training and testing of the FVMD task, networks were first presented with two color targets through the target input units, and then presented with the motion direction through the motion input units. Both the motion and target inputs persisted until the end of each trial. A short time after the motion input (100ms, 5 time steps), the networks were required to report the direction of the stimulus motion by choosing the saccade target with the appropriate color. Specifically, each element of the task design in the FVMD task that the models were trained to perform was tailored to match those used in the monkey experiments: the directions of the motion stimuli, the target colors, as well as the task (stimulus) durations were the same as those used in monkey electrophysiology experiments. Trials were programmatically generated by constructing motion/target inputs to the networks at each timestep and the desired response (left saccade, right saccade) at each timestep. In total, we trained 50 networks to perform the FVDM task. All networks were first trained using motion directions with two coherence levels (60% and 90%) and were then tested using motion stimuli with another four different coherence levels (75%, 55%, 35%, and zero), which had never been presented during training. Besides the difference in the coherence of motion input, the color inputs remained the same during the testing period. All the networks achieved consistently high performance by the end of training (>99% accuracy). All the important model hyperparameters were listed in *Supplementary file 1*.

## Quantification of the networks' behavioral performance

To test whether our multi-module RNNs exhibited decision behavior similar to monkey subjects, we examined their performance accuracies and RTs when tested by novel stimuli with different coherence levels. The performance accuracy was defined as whether the output of the response units (starting from 100ms after stimulus onset to the trial end) matched the desired output. Furthermore, we defined the networks' RTs as the time point from which the differences between two output units were greater than a threshold (0.8) for three consecutive time points after the stimulus onset. During some trials, the threshold was never reached until the end of the trial (500ms after the motion stimulus onset). In these cases, we artificially set the RT to be 600ms in these trials for further analysis.

## Analysis of RNN activity

We performed the same analyses on the RNN units as we did on the neurophysiology data. In *Figures 4–5*, data from both the example network and averaged results across all the networks were shown. For every network, we only included the units which exhibited at least one kind of task-related modulation during the test period (i.e. motion DS or saccade DS, one-way ANOVA test, p<0.01). In order to quantify the activity related to evaluating motion stimulus, we analyzed the motion DS only including the units in the motion modules, whereas only the saccade DS of units in the target modules was calculated to quantify the activity related to saccade selection during the decision-making.

## Analysis of RNN connectivity

To examine the potential circuit mechanisms underlying the modulation of action selection on sensory evaluation during decision-making in the RNNs, we examined the cross-module connection weights in the trained networks. Specifically, we defined the feedforward connection as the projection from units in the motion module to units in the target module within the same main module (hemisphere). We also defined the feedback connection as the projection from units in the target module to units in the motion module within the same main module. Furthermore, we grouped the cross-module connections into the selectivity-matched and selectivity-mismatched groups based on the preferences of the neural encoding of units in the motion and target modules. The selectivity-matched group included two types of unit pairs: (1) units in the motion module preferred to encode 315° and units in the target module preferred to encode red, (2) units in the motion module preferred to encode 135° and units in the target module preferred to encode green. The selectivity-mismatched group includes the other two types of unit pairs: (3) units in the motion module preferred to encode 315° while units in the target module preferred to encode green, and (4) units in the motion module preferred to encode 135° while units in the target module preferred to encode red. The values of connection weights projected from excitatory and inhibitory units were set as positive and negative, respectively.

We also examined the correlation between the similarity of neural selectivity and connection strength for different unit pairs in the RNNs based on the following steps: First, both the preference

and strength of motion direction encoding or target color encoding were quantified for units in the motion module or target module, respectively. This was done by calculating the averaged differential activity to the 315° and 135° motion stimuli after motion stimulus onset or calculating the averaged differential activity to the red and green targets after target onset. In this case, the unit pairs that exhibited matched selectivity would show selectivity values with the same sign (positive or negative), whereas the unit pairs that exhibited mismatched selectivity would show selectivity values with the opposite signs. Second, the units in both the motion module and target modules were ranked based on selectivity values. Third, the selectivity similarity of each cross-module unit pair was quantified by calculating the reverse value of the absolute differences between the target color selectivity rank and the motion DS rank. The rank of the weight value of each pair was calculated and then reversed to give the lower weight value a higher rank score. Finally, we calculated the Pearson correlation between weight rank and selectivity similarity. A positive correlation indicates that units that exhibited stronger encoding preference to one motion direction in the motion module were more likely to connect with the units in the target module that exhibited stronger encoding on the matched target color.

## Inactivation experiments in silico

Our examination of the network connectivity within the hidden layer of the RNNs revealed that the selectivity-specific top-down connections originated from the target module to the motion module. To assess the causal contribution of such precise feedback connections to the modulation of action selection on sensory evaluation, we performed two in silico analogues of neuronal inactivation experiments similar to a previous study (*Zhou et al., 2021*). We hypothesized that ablating or disrupting the precise feedback connection from the target module to the motion module would significantly impact the activity patterns of the units in the motion module. Specifically, such inactivation on connectivity was expected to reduce the difference in motion DS in the notion module between CT and IT conditions. To test this hypothesis, we performed the projection-specific inactivation experiments in the RNNs in three different ways. First, we ablated all the feedback projections from the target module to the motion module while keeping all the other network parameters unchanged after the networks were fully trained. We then tested the networks with the untrained motion stimuli used in the normal experiment. The connectivity ablation was implemented by directly setting the connection weights from units in the target module to units in the motion module to zero. Second, we randomly rearranged the connection weights from the target module to the motion module after the network was fully trained by shuffling the corresponding weights in the weight matrix. Then, we tested the networks with the testing motion stimuli. This was aimed to disrupt the selectivity-specific feedback connection without changing the total strength of the feedback connection. We repeated this random feedback weight rearrangement process 100 times for each of the 50 trained networks. Subsequently, we measured the impact on the networks' behavior performance and the activity patterns in both the motion and saccade modules. In particular, for each network, we tested whether there were still different levels of motion DS of units in the motion module between CT and IT conditions after inactivating the feedback connectivity.

We further performed pattern-specific ablation of the feedback connections in the RNNs, in order to examine the causal roles of different patterns of feedback connectivity in solving the FVDM task. Specifically, we separated the feedback projections in each RNN into the specific group and the nonspecific group. The specific group included two conditions: (1) the feedback connection weight was positive between the selectivity-matched unit pairs and (2) the feedback connection weight was negative between the selectivity-nonmatched unit pairs, whereas the nonspecific group included the rest of the feedback conditions in each RNN. We then tested the effects of the pattern-specific connectivity inactivation on the RNNs' behavior performance when tested with the untrained motion stimuli.

In the third inactivation experiment, we trained 50 additional RNNs without feedback connections to learn the FVDM task, in order to test the importance of feedback connectivity in solving the flexible decision task. These networks were initialized using identical hyperparameters and trained with the same methodology as the normal RNNs, except that the feedback connections from the target module to the motion module were ablated before training. We then examined these RNNs' behavior performance and unit activity when testing with the untrained motion stimuli used in the normal experiment. Furthermore, we trained another 50 RNNs without feedback connection to

learn the FVMD task. These RNNs were initialized with higher recurrent connection probabilities than the full-model RNNs to keep the number of the total trainable connection weight comparable to that in the full-model RNNs. We raised the recurrent connection probability by a factor of 1.176 after removing feedback connections (average number of connections in 50 normal RNNs = 8193, average number of connections in 50 networks without feedback = 8184, p=0.53, independent sample t-test). Except for the recurrent connection probability, all the other hyperparameters and the training methodology were the same as the normal RNNs. We then compared the behavior performance of these RNNs with that of the normal RNNs when testing with the untrained motion stimuli.

## Analysis of neural landscape

For each network, we adopted a methodology similar to a previous study (cite) to reconstruct the energy landscape of population activity within the target module. Initially, units in the target module exhibiting saccade DS were identified. Subsequently, we employed principal component analysis (PCA) to reduce the dimensionality of the population activity. Specifically, PCA was performed on the average activity across these units, considering each stimulus motion direction, coherence level, and choice. We retained the first five principal components (PCs) for further analysis, as they explained over 90% of the total variance in population activity. Following this, the trial-by-trial population activities were transformed into five-dimensional data using the aforementioned five PCs.

To predict trial-by-trial choice, we employed a linear Support Vector Machine (SVM) classifier with 10-fold cross-validation. Specifically, we utilized the average activity of the last 100ms of each trial as input for the classifier. The choice axis was determined by selecting the normal vector of the separating hyperplane from the SVM classifier that demonstrated the best performance. Subsequently, the five-dimensional data points were projected onto the choice axis, resulting in one-dimensional projections. These projections represented the positions along the choice axis, with the intersection between the normal vector and the hyperplane serving as the zero point.

The potential in this context is determined by integrating the spatial component of the time derivative of the population activity, denoted as $X_t$, for a given condition. To be more precise, we began by computing the expected value across trials of the time derivative of the unit activities for each choice. Subsequently, we employed the potential function $V(x, t)$ to evaluate the potential at position x and time t,

$$V(x, t) = \left( \sum_{p < x} \frac{dX_t}{dt} | X_t = p \right) \cdot (-\Delta x)$$

Crucially, we computed the potential separately for each choice, establishing the potential at position 0 as the zero-potential reference. To provide a holistic evaluation of the potential landscape, we combined the potentials of both choices at position 0.

Furthermore, we expanded the computation of potentials to encompass each stimulus coherence level using the aforementioned approach. Similar to the prior analysis, the time derivatives were still derived from the 1-D projections of each data point onto the choice axis. However, in this instance, the conditions were based on the stimulus coherence level rather than the choice. For a representative potential value for each model, we averaged the potentials at the time point of 300ms after the stimulus onset. Additionally, recognizing that the positions visited by each model might differ, we ensured the accuracy of standard error estimation by retaining only the potential values at positions continuously visited by four or more models.

## Acknowledgements

We thank Tianming Yang, Matthew Rosen, Siyu Wang, Cheng Xue, Gongcheng Yu, Mingze Li and Ou Zhu for their comments on an earlier version of this manuscript. We also thank the veterinary staff at The University of Chicago Animal Resources Center for expert assistance. This study was supported by STI2030-Major Projects (2021ZD0203800), NSFC32171036.

# Additional information

## Funding

| Funder | Grant reference number | Author |
|---|---|---|
| Ministry of Science and Technology of the People's Republic of China | STI2030-Major Projects (2021ZD0203800) | Yang Zhou |
| National Natural Science Foundation of China | NSFC32171036 | Yang Zhou |

The funders had no role in study design, data collection and interpretation, or the decision to submit the work for publication.

## Author contributions

Xuanyu Wu, Formal analysis, Visualization, Methodology, Writing – review and editing; Yang Zhou, Conceptualization, Resources, Data curation, Formal analysis, Supervision, Funding acquisition, Validation, Investigation, Visualization, Methodology, Writing - original draft, Project administration, Writing – review and editing

## Author ORCIDs

Xuanyu Wu (iD) https://orcid.org/0009-0009-8607-9381
Yang Zhou (iD) https://orcid.org/0000-0002-4517-1052

## Ethics

All experimental and surgical procedures were performed in strict accordance with the recommendations in the Guide for the Care and Use of Laboratory Animals of the National Institutes of Health. All of the animals were handled according to approved institutional animal care and use committee (IACUC) protocol #71887 of The University of Chicago.

Reviewer #2 (Public review): https://doi.org/10.7554/eLife.96402.3.sa1
Author response https://doi.org/10.7554/eLife.96402.3.sa2

# Additional files

## Supplementary files

MDAR checklist

Supplementary file 1. Table S1.

## Data availability

The electrophysiology data is available on Dryad. The code for Training RNN and the relative analysis is available on GitHub (copy archived at *Wu, 2025*).

The following dataset was generated:

| Author(s) | Year | Dataset title | Dataset URL | Database and Identifier |
|---|---|---|---|---|
| Wu X, Zhou Y | 2025 | Recorded neural data from: Nonlinear feedback modulation contributes to the optimization of flexible decision-making | https://doi.org/10.5061/dryad.n8pk0p37z | Dryad Digital Repository, 10.5061/dryad.n8pk0p37z |

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
