## [Editor Report · eLife Assessment]

This **valuable** study by Wu and Zhou combines neurophysiological recordings and computational modelling to address an interesting question regarding the sequence of events from sensing to action. Neurophysiological evidence remains **incomplete**: explicit mapping of saccade-related activity in the same neurons and a better understanding of the influence of the spatial configuration of stimulus and targets would be required to pinpoint whether such activity might contribute, even partially, to the observed results and interpretations. These results are of interest for neuroscientists investigating decision-making.

---

## [Referee Report · Reviewer #2 (Public review)]

Summary:

In this manuscript, the authors recorded activity in the posterior parietal cortex (PPC) of monkeys performing a perceptual decision-making task. The monkeys were first shown two choice dots of two different colors. Then, they saw a random dot motion stimulus. They had to learn to categorize the direction of motion as referring to either the right or left dot. However, the rule was based on the color of the dot and not its location. So, the red dot could either be to the right or left, but the rule itself remained the same. It is known from past work that PPC neurons would code the learned categorization. Here, the authors showed that the categorization signal depended on whether the executed saccade was in the same hemifield as the recorded PPC neuron or in the opposite one. That is, if a neuron categorized the two motion directions such that it responded stronger for one than the other, then this differential motion direction coding effect was amplified if the subsequent choice saccade was in the same hemifield. The authors then built a computational RNN to replicate the results and make further tests by simulated "lesions".

Strengths:

Linking the results to RNN simulations and simulated lesions.

Weaknesses:

Potential interpretational issues due to a lack of explicit evidence on the sizes and locations of the response fields of the neurons. For example, is the contra/ipsi effect explained by the fact that in the contra condition, the response target and the saccade might have infringed on the outer edges of the response fields?

---

## [Author Response]

The following is the authors’ response to the original reviews

**Reviewer #1 (Public Review):**
Summary:This valuable study by Wu and Zhou combined neurophysiological recordings and computational modelling to investigate the neural mechanisms that underpin the interaction between sensory evaluation and action selection. The neurophysiological results suggest non-linear modulation of decision-related LIP activity by action selection, but some further analysis would be helpful in order to understand whether these results can be generalised to LIP circuitry or might be dependent on specific spatial task configurations. The authors present solid computational evidence that this might be due to projections from choice target representations. These results are of interest for neuroscientists investigating decision-making.Strengths:Wu and Zhou combine awake behaving neurophysiology for a sophisticated, flexible visual-motion discrimination task and a recurrent network model to disentangle the contribution of sensory evaluation and action selection to LIP firing patterns. The correct saccade response direction for preferred motion direction choices is randomly interleaved between contralateral and ipsilateral response targets, which allows the dissociation of perceptual choice from saccade direction.The neurophysiological recordings from area LIP indicate non-linear interaction between motion categorisation decisions and saccade choice direction.The careful investigation of a recurrent network model suggests that feedback from choice target representations to an earlier sensory evaluation stage might be the source for this non-linear modulation and that it is an important circuit component for behavioural performance.The paper presents a possible solution to a central controversy about the role of LIP in perceptual decision-making, but see below.Weaknesses:The paper presents a possible solution to a central controversy about the role of LIP in perceptual decision-making. However, the authors could be more clear and upfront about their interpretational framework and potential alternative interpretations.Centrally, the authors' model and experimental data appears to test only that LIP carries out sensory evaluation in its RFs. The model explicitly parks the representation of choice targets outside the "LIP" module receiving sensory input. The feedback from this separate target representation provides then the non-linear modulation that matches the neurophysiology. However, they ignore the neurophysiological results that LIP neurons can also represent motor planning to a saccade target.The neurophysiological results with a modulation of the direction tuning by choice direction (contralateral vs ipsilateral) are intriguing. However, the evaluation of the neurophysiological results are difficult, because some of the necessary information is missing to exclude alternative explanations. It would be good to see the actual distributions and sizes of the RF, which were determined based on visual responses not with a delayed saccade task. There might be for example a simple spatial configuration, for example, RF and preferred choice target in the same (contralateral) hemifield, for which there is an increase in firing. It is a shame that we do not see what these neurons would do if only a choice target would be put in the RF, as has been done in so many previous LIP experiments. The authors exclude also some spatial task configurations (vertical direction decisions), which makes it difficult to judge whether these data and models can be generalised. The whole section is difficult to follow, partly also because it appears to mix reporting results with interpretation (e.g. "feedback").The model and its investigation is very interesting and thorough, but given the neurophysiological literature on LIP, it is not clear that the target module would need to be in a separate brain area, but could be local circuitry within LIP between different neuron types.
**Reviewer #2 (Public Review):**
Summary:In this manuscript, the authors recorded activity in the posterior parietal cortex (PPC) of monkeys performing a perceptual decision-making task. The monkeys were first shown two choice dots of two different colors. Then, they saw a random dot motion stimulus. They had to learn to categorize the direction of motion as referring to either the right or left dot. However, the rule was based on the color of the dot and not its location. So, the red dot could either be to the right or left, but the rule itself remained the same. It is known from past work that PPC neurons would code the learned categorization. Here, the authors showed that the categorization signal depended on whether the executed saccade was in the same hemifield as the recorded PPC neuron or in the opposite one. That is, if a neuron categorized the two motion directions such that it responded stronger for one than the other, then this differential motion direction coding effect was amplified if the subsequent choice saccade was in the same hemifield. The authors then built a computational RNN to replicate the results and make further tests by simulated "lesions".Strengths:Linking the results to RNN simulations and simulated lesions.Weaknesses:Potential interpretational issues due to a lack of evidence on what happens at the time of the saccades.
**Recommendations for the authors:**

**Reviewer #1 (Recommendations For The Authors):**
(1) The neurophysiological results with a modulation of the direction tuning by choice direction are intriguing. However, the evaluation of the neurophysiological results are difficult because some of the necessary information is missing to exclude alternative explanations.

We thank the reviewer for the helpful comments. We have addressed this point in detail in the following response.

(a) Clearly state in the results how the response field "RF", where the stimulus was placed, was mapped. The methods give as "MGS"" (i.e., spatial selectivity during stimulus presentation and delay)" task rather than the standard delayed saccade. And also "while for those neurons which did not show a clear RF during the MGS task, we presented motion stimuli in the positions (always in the visual field contralateral to the recorded hemisphere) in which neurons exhibited the strongest response to the motion stimuli." All this sounds more like a sensory receptive field not an eye movement response filed". What was the exact task and criterion?

We agree with the reviewer that the original description of how we mapped the response fields (RFs) of LIP neurons lacked sufficient detail. In this study, we used the memory-guided saccade (MGS) task to map the RFs of all isolated LIP neurons. Both MGS and delayed saccade tasks are commonly used to map a neuron's response field in previous decision-making studies.

In the MGS task, monkeys initially fixate on the center of the screen. Subsequently, a dot randomly flashes at one of the eight possible locations surrounding the fixation dot with an eccentricity of 8 degree, requiring the monkeys to memorize the location of the flashed dot. After a delay of 1000 ms, the monkeys are instructed to saccade to the remembered location once the fixation dot disappears. The MGS task is a standard behavior task for mapping visual, memory, and motor RFs, particularly in brain regions involved in eye movement planning and control, such as LIP, FEF, and the superior colliculus.

We believe the reviewer's confusion may stem from whether we mapped the visual, memory, or motor RFs of LIP neurons in the current study, as these "RFs" are not always consistent across individual neurons. In our study, we primarily mapped the visual and memory RFs of each LIP neuron by analyzing their activity during both the target presentation and delay periods. To focus on sensory evaluation-related activity, we presented the visual motion stimulus within the visual-memory RF of each neuron. For neurons that did not show a significant visual-memory RF, we used a different approach: we tested the neurons with the main task by altering the spatial configuration of the task stimuli to identify the visual field that elicited the strongest response when the motion stimulus was presented within it. This approach was used to guide the placement of the stimulus during the recording sessions.

Following the reviewer’s suggestion, we have added the following clarification to the results section to better describe how we mapped the RF of LIP neurons:

‘We used the memory-guided saccade (MGS) task, which is commonly employed in LIP studies, to map the receptive fields (RFs) of all isolated LIP neurons. Specifically, we mapped both the visual and memory RFs of each neuron by analyzing their activity during the target presentation and delay periods of the MGS task (see Methods).’.

(b) l.85 / l126: What do you mean by "orthogonal to the axis of the neural RF" - was the RF shape asymmetric, if so how did you determine this? OR do you mean the motion direction axis? Please explain.

We realized that the original description of this point may have been unclear and could lead to confusion. The axis of the neural RF refers to the line connecting the center of the RF (which coincides with the center of the motion stimulus) to the fixation dot. We have revised this sentence in the revised manuscript as follows:

‘To examine the neural activity related to the evaluation of stimulus motion, we presented the motion stimuli within the RF of each neuron, while positioning the saccade targets at locations orthogonal to the line connecting the center of the RF (which also marks the center of the motion stimulus) and the fixation dot.’

(c) Behavioural task. Figure 1 - are these example session? Please state this clearly. Can you show the examples (psychometric function and reaction times) separated for trials where correct choice direction aligning with the motion preference (within 90 degrees) and those that did not?

Figure 1 shows the averaged behavioral results from all recording sessions. We have added this detail in the revised legend of Figure 1.

We are uncertain about the reviewer’s reference to the “correct choice direction aligning with the motion preference,” as the term “motion preference” is specific to the neuron response, which are different for different neurons recorded simultaneously using multichannel recording probe.

Nonetheless, following the reviewer’s suggestion, we grouped the trials in each recording session into two groups based on the relationship between the saccade direction and the preferred motion direction of the identified LIP neuron during one example single-channel recording. Both the RT and the performance accuracy during one example session were shown in the following figure.

**Author response image 1. sa2fig1:** 

Give also the performance averaged across all sites included in this study and range.If performance does differ for different configuration, please, show that the main modulatory effect does not align with this distinction.

To clarify this point, we have plotted performance accuracy and RTs for horizontal, oblique, and vertical target position configurations separately, which are shown for both monkeys in the following figures. We did not observe any systematic influences of task configurations on the monkeys' performance accuracy. While the RTs did differ across different configurations, we believe these differences are likely attributable to several factors, such as varying levels of familiarity introduced by our training process and the intrinsic RT difference between different saccade directions.

(d) Show the distribution of RF positions and the direction preferences for the recording sites included in the quantitative analysis of this study. (And if available, separately those excluded).

Following the reviewer’s suggestion, we have plotted the centers of the RFs for all neurons with identifiable RFs, categorizing them by their preferred motion directions. To determine each neuron’s RF, we analyzed the average firing rates from both the target presentation and delay periods during each trial of the memory-guided saccade (MGS) task. The RF centers of neurons with significant RFs were determined through a two-step process. First, we selected neurons that exhibited significant RFs in the MGS based on the following criteria: (1) there must be a significant activity difference between the eight target locations, and (2) the mean activity during the selected periods should be significantly greater than the baseline activity during the fixation period. Second, we fitted the activity data from the eight conditions to a Gaussian distribution, using the center of the fitted distribution as the RF center. A significant proportion of neurons from both monkeys that exhibited significant response to motion stimuli did not exhibited notable RFs based our current method. The following figures show the distributions of RFs and motion direction preference for all LIP neurons with identifiable RFs separately for each monkey. Since this is not the focus of the current study, we are not planning to include this result in the revised manuscript.

**Author response image 3. sa2fig3:** 

(e) Following on from (d), was there a systematic relationship between RF position or direction preference and modulation by choice direction? For instance could the responses be simply explained by an increase in modulation for choices into the same (contralateral) hemifield as where the stimulus was placed?

The reviewer raised a good point. To address whether there was a systematic relationship between RF position or direction preference and modulation by choice direction, we calculated a modulation index for each neuron to quantify the influence of saccade direction on neuronal responses to motion stimuli. We then plotted the modulation index against the RF position for each LIP neuron, shown as following:

**Author response image 4. sa2fig4:** 

As shown in the figures above, neurons with RFs farther from the horizontal meridian were more likely to exhibit stronger modulation by the saccade direction, while neurons with RFs closer to the horizontal meridian showed inconsistent and weaker modulation. This is because when the RFs was on the horizontal meridian, saccade directions were aligned with the vertical axis (with no contralateral or ipsilateral directions). This is consistent with the finding in Figure S3—no significant differences in direction selectivity between the CT and IT conditions in the data sessions where the saccade targets were aligned close to the vertical direction. Since fewer than half of the identified neurons showed clear receptive fields using our method, the figure above did not include all the neurons used in the analysis in the manuscript. Therefore, we chose not to include this figure in the revised manuscript.

Additionally, we quantified the relationship between the modulation index and direction preference for neurons in sessions where the monkeys’ saccades were aligned to either horizontal or oblique directions. As shown in the following figure, no systematic relationship was found between direction preference and modulation by the choice direction for LIP neurons at the population level.

**Author response image 5. sa2fig5:** 

We have added this result as Figure S 2 in the revised manuscript.

Notably, the observed modulation of saccade direction on LIP neurons’ response to motion stimuli cannot be simply explained by saccade direction selectivity. We presented two more evidence to rule out such possibility in the original manuscript. First, the modulation effect we observed was nonlinear; specifically, the firing rate of neurons increased for the preferred motion direction but decreased for the non-preferred motion direction (Figure 2i and Figure S1A-D). This phenomenon is unlikely to be attributed to a linear gain modulation driven by saccade directions. Second, we plotted the averaged neural activity for contralateral and ipsilateral saccade directions separately, and found that LIP neurons showed similar levels of activity between two saccade directions (revised Figure 2L).

Additionally, we added a paragraph in the Methods section to describe the way we calculated modulation index as follows:

“We have calculated a modulation index for each neuron to reflect the influence of saccade direction on neuron’s response to visual stimuli. The modulation index is calculated as:\begin{document}$$\displaystyle \delta_{\text {contra }}=r_{\text {pref }}^{\text {contra }}-r_{\text {non-pref }}^{\text {contra }}$$\end{document}\begin{document}$$\displaystyle \delta_{i p s i}=r_{p r e f}^{i p s i}-r_{n o n-p r e f}^{i p s i}$$\end{document}\begin{document}$$\displaystyle \mathrm{MI}=\frac{\delta_{\text {contra }}-\delta_{\mathrm{ipsi}}}{\delta_{\text {contra }}+\delta_{\mathrm{ipsi}}}$$\end{document}

where \begin{document}$r_{\text {pref }}^{\text {contra }}$\end{document} represents the average firing rate from 50ms to 250ms after sample onset for all contralateral saccade trails with a neuron’s preferred moving direction of visual stimuli. The naming conventions are the same for \begin{document}$r_{\text {non-pref }}^{\text {contra }}$\end{document}, \begin{document}$r_{\text {pref }}^{i p s i}$\end{document}, and \begin{document}$r_{\text {non-pref }}^{\text {ipsi }}$\end{document}. An MI value between 0 and 1 indicate higher modulation in contralateral saccade trials, and an MI value between -1 and 0 indicates higher modulation in ipsilateral saccade trials.”

Please split Figures 2G,H,I J,K, by whether the RF was located contralaterally or ipsilaterally. If there are only a small number of ipsilateral RFs, please show these examples, perhaps in an appendix.

This is a reasonable suggestion; however, it is not applicable to our study. Among all the neurons included in our analysis, only one neuron from each monkey exhibited ipsilateral receptive fields (RFs). Therefore, we believe it may not be necessary to plot the result for this outlier.

(f) Were the choice targets always equi-distant from the stimulus and at what distance was this? Please give quantitative details in methods.

The review was correct that the choice targets were always equidistant form the stimulus. The distance between the motion stimulus and the target was typically 12-15 degree. We have added the details in the revised Methods section as follows:

‘Therefore, the two saccade targets were equidistant from the stimulus, with the distance typically ranging from 12 to 15 degrees.

(2) For Figure 3E, how do you explain that there is an up regulation of for contralateral choices before the stimulus onset, i.e. before the animal can make a decision? Is this difference larger for error trials?

This is a good question, which we have attempted to clarify in the revised manuscript. We believe that the observed upregulation in neural activity for contralateral choices may reflect the monkeys’ internal choice bias or expectation (choice between two motion directions) prior to stimulus presentation, which could influence their subsequent decisions. In Figure 3E, we calculated the r-choice to assess the correlation between the neuron’s direction selectivity and the monkeys’ decisions on motion stimuli, separately for contralateral and ipsilateral choice conditions. The increased r-decision during the pre-stimulus period indicates stronger neural activity for trials in which the monkeys later reported that the upcoming stimulus was in the preferred direction, and weaker activity for trials where the stimulus was judged to be in the non-preferred direction. This correlation was more pronounced for contralateral choices than for ipsilateral ones. It is important to note that while the monkeys cannot predict the upcoming stimulus direction with greater-than-chance accuracy, these results suggest that pre-stimulus neural activity in LIP is correlated with the monkeys’ eventual decision for that trial. Furthermore, LIP neural activity was more strongly correlated with the monkeys’ decisions in the contralateral choice condition compared to the ipsilateral one.

Additionally, we clarify that the r-decision was calculated using both correct and error trials. When comparing Figure 2J with Figure 2K, the correlation between neural activity and the monkeys’ upcoming decision during the pre-stimulus period was most prominent in low- and zero-coherence trials, where the monkeys either made more errors or based decisions on guesswork. We infer that the monkeys' confidence in these decisions was likely lower compared to high-coherence trials. Thus, the decision process appears to be influenced by pre-stimulus neural activity, particularly in low-coherence and zero-coherence trials.

Although it is unclear precisely what covert process this pre-stimulus activity reflects, similar patterns of choice-predictive pre-stimulus activity have been observed in LIP and other brain areas (Shadlen, M.N. and Newsome,T.W., 2001; Coe, B., at al. 2002; Baso, M.A. and Wurtz, R.H., 1998; Z. M. Williams at al. 2003). We have clarified this point in the revised manuscript, including a revision of the relevant sentence in the Results section for clarity, shown as follows:

“Furthermore, we used partial correlation analysis to examine decision- and stimulus-related components of DS (i.e., r-decision and r-stimulus, Figure 3E and 3F) using all four coherence levels. The decision-related component of LIP DS was significantly greater in the CT condition than in the IT condition (Figure 3E; nested ANOVA: P = 1.07e-6, F = 25.72), and this difference emerged even before motion stimulus onset. This suggests that the LIP DS was more closely correlated with monkeys’ decisions in the CT condition than in the IT condition. The upregulation in r-decision for contralateral choices may reflect the monkeys’ internal choice bias or expectation (choice between two motion directions) prior to stimulus presentation, which could influence their subsequent decisions more in the CT condition”

(3) Figure 2K: what is the very large condition-independent contribution? It almost seems as most of what these neurons code for is neither saccade or motion related.

The condition-independent contribution is the time-dependent component that is unrelated to saccade, motion, or their interaction. Our findings are consistent with previous methodological studies, where this time-dependent component was shown to account for a significant portion of the variance in population activity (Kobak, D. et al., 2016)

(4) Abstract:a) "We found that the PPC activity related to monkeys' abstract decisions about visual stimuli was nonlinearly modulated by monkeys' following saccade choices directing outside each neuron's response field."This sentence is not clear/precise in two regards:Should "directing" be "directed"?Also, it is not just saccades directed outside the RF, but towards the contralateral hemifield.

We thank the reviewer for the suggestion. We agree that ‘directing’ should be ‘directed’ and revised it accordingly. However, we do not believe that ‘directed outside each neuron's response field’ should be replaced with “towards the contralateral hemifield”. There are two major reasons. First, the modulation effect was identified as the difference between contralateral and ipsilateral saccade directions. We cannot conclude that the modulation mainly happened in the contralateral saccade direction. Second, we used ‘directed outside each neuron's response field’ to emphasize that this modulation cannot be simply explained by saccade direction selectivity, whereas ‘towards the contralateral hemifield’ cannot fulfill this purpose.

(b) " Recurrent neural network modeling indicated that the feedback connections, matching the learned stimuli-response associations during the task, mediated such feedback modulation."- should be "that feedback connection .... might mediate". A model can only ever give a possible explanation.

Thanks for the help on the writing again! We have revised this sentence as following: “Recurrent neural network modeling indicated that the feedback connections, matching the learned stimuli-response associations during the task, might mediate such feedback modulation.”

(c) "thereby increasing the consistency of flexible decisions." I am not sure what is really meant by increasing the consistency of flexible decisions? More correct or more the same?

We apologize for the confusion. In the manuscript, "decision consistency" refers to the degree of agreement in the model's decisions under specific conditions. A higher decision consistency indicates that the model is more likely to produce the same choice when encountering encounters a stimulus in that condition. We have incorporated your suggestion and revise this sentence as “thereby increasing the reliability of flexible decisions”. We also clarified the definition of consistency in the main text as follows:

“These disrupted patterns of saccade DS observed in the target module following projection-specific inactivation aligned with the decreased decision consistency of RNNs, where decision consistency reflects the degree of agreement in the model's choices under specific task conditions. This suggests a diminished reliance on sensory input and an increased dependence on internal noise in the decision-making process.”.

(5) Results: headers should be changed to reflect the actual results, not the interpretation:"Nonlinear feedback modulation of saccade choice on visual motion selectivity in LIP""Feedback modulation specifically impacted the decision-correlated activity in LIP"These first parts of the results describe neurophysiological modulations of LIP activity, the source cannot be known from the presented data alone. I thought that this feedback is suggested by the modelling results in the last part of the results. It is confusing to the reader that the titles already refer to the source of the modulation as "feedback". The titles should more accurelty describe what is found, not pre-judge the interpretation.

We thank the reviewer for those valuable suggestions. We have updated the subtitles to: “Nonlinear modulation of saccade choice on visual motion selectivity in LIP” and “Decision-correlated but not stimulus-correlated activity was modulated in LIP.”

(6) page 8, l366-380. Can you link the statements more directly to panels in Figure 6. For Figure 6H-K, it needs to be clarified that the headers for 6D-G also apply to H-K.

­We have added headers for Figure 6H-K in the revised version, and revised the corresponding results section as follows.

‘We further examined how the energy landscape in the 1-D subspace changed in relation to task difficulty (motion coherence). Consistent with prior findings, trials with lower decision consistency (trials using lower motion coherence) exhibited shallower attractor basins at the time of decision for all types of RNNs (Fig. 6H-K). However, both the depth and the positional separation of attractor basins in the network dynamics significantly decreased for all non-zero motion coherence levels after the ablation of all feedback connections (comparing Figure 6I with Figure 6H; P(depth) = 5.20e-25, F = 122.80; P(position) = 1.82e-27, F = 137.75; two-way ANOVA). Notably, this reduction in basin depth and separation was more pronounced in the specific group compared to the nonspecific groups after ablating the feedback connections (comparing Figure 6J with Figure 6K; P(depth) = 2.65e-13, F = 57.35; P(position) = 3.73e-14, F = 61.79; two-way ANOVA). These results might underlie the computational mechanisms that explain the observed reduction in the decision consistency of RNNs following projection-specific inactivation: the shallower and closer attractor basins after ablating feedback connections resulted in less consistent decisions. This happened because the variability in neural activity made it more likely for population activity to stochastically shift out of the shallower basins and into nearby alternative ones.’

(7) line 556-557: Please provide a reference or data for the assertion that nearby recording sites in LIP (100 microns apart) have similar RFs.

The reviewer raised an interesting question that we are unable to address in depth with the current data, as we lack information on the specific cortical location for each recording session. In the original manuscript, we suggested that nearby recording sites in LIP have similar receptive fields (RFs), based on both our own experience with LIP recordings and previous studies. Specifically, we observed that neurons recorded within a single penetration using a single-channel electrode typically exhibited similar RFs. Similarly, the majority of neurons recorded from the same multichannel linear probe within a single session also showed comparable RFs. Additionally, several studies (both electrophysiological and fMRI) have reported topographic organization of RFs in LIP (Gaurav H. Patel et al., 2010; S. Ben Hamed et al., 2001; Gene J. Blatt et al., 1990).

(8) Line 568, Methods: a response criterion of a maximum firing rate of 2 spikes/s seems very low, especially for LIP. How do the results change if this lifted to something more realistic like 5 spikes/s or 10 spikes/s?

We chose this criterion to ensure we included as many neurons as possible in our analysis. To further clarify, we have plotted the distribution of maximum firing rates across all neurons. Based on our findings, relaxing this criterion is unlikely to affect the results, as the majority of neurons exhibit maximum firing rates well above 5 spikes/s, and many exceed 10 spikes/s. We hope this explanation addresses the concern.

**Author response image 6. sa2fig6:** 

**Reviewer #2 (Recommendations For The Authors):**
In this manuscript, the authors recorded activity in the posterior parietal cortex (PPC) of monkeys performing a perceptual decision-making task. The monkeys were first shown two choice dots of two different colors. Then, they saw a random dot motion stimulus. They had to learn to categorize the direction of motion as referring to either the right or left dot. However, the rule was based on the color of the dot and not its location. So, the red dot could either be to the right or left, but the rule itself remained the same. It is known from past work that PPC neurons would code the learned categorization. Here, the authors showed that the categorization signal depended on whether the executed saccade was in the same hemifield as the recorded PPC neuron or in the opposite one. That is, if a neuron categorized the two motion directions such that it responded stronger for one than the other, then this differential motion direction coding effect was amplified if the subsequent choice saccade was in the same hemifield. The authors then built a computational RNN to replicate the results and make further tests by simulated "lesions".The data are generally interesting, and the manuscript is generally well written (but see some specific comments below on where I was confused). However, I'm still not sure about the conclusions. The way the experiment is setup, the "contra" saccade target is essentially in the same hemifield as the motion patch stimulus. Given that the RF's can be quite large, isn't it important to try to check whether the saccade itself contributed to the effects? i.e. if the RF is on the left side, and the "contra" saccade is to the left, then even if it is orthogonal to the location of the stimulus motion patch itself, couldn't the saccade still be part of a residual edge of the RF? This could potentially contribute to elevating the firing rate on the preferred motion direction trials. I think it would help to align the data on saccade onset to see what happens. It would also help to have fully mapped the neurons' movement fields by asking the monkeys to generate saccades to all screen locations in the monitor. The authors mention briefly that they used a memory-guided saccade task to map RF's, but it is also important to map with a visual target. And, in any case, it would be important to show the mapping results aligned on saccade onset.Another comment is that the authors might want to mention this other recent related paper by the Pack group: https://www.biorxiv.org/content/10.1101/2023.08.03.551852v2.full.pdf

We thank the reviewer for the comments and realized that we did not explain our results clearly in the original manuscript. We agree with the reviewer that saccade direction selectivity might be a confounding factor for the modulation of the saccade choice direction onto LIP neurons’ activity responded to visual motion stimuli. Because the RFs of LIP neurons might be large and the saccade target might be presented within the edge of the RFs. However, we believe that the observed modulation of saccade direction on LIP neurons’ response to motion stimuli cannot be simply explained by saccade direction selectivity. We presented several pieces of evidence to rule out such possibility. First, the modulation effect we observed was not linear; specifically, the firing rate of neurons increased for the preferred motion direction but decreased for the non-preferred motion direction (Figure 2i and Figure S1A-D). This phenomenon is unlikely to be attributed to a linear gain modulation driven by saccade directions. Second, we plotted the averaged neural activity for contralateral and ipsilateral saccade directions separately, aligned the activity to either motion stimulus onset or saccade onset, and found that LIP neurons showed similar levels of activity between the contralateral and ipsilateral directions (revised Figure 2L), which is not consistent with obvious saccade direction selectivity.

To better control for this confound, we have added figures plotting the mean neural activity aligned to saccade onset for both contralateral and ipsilateral saccades, which are now included in the revised main Figure 2. These figures are presented in the detailed response below. Additionally, we have revised the corresponding results section to clarify our points, as outlined below:

“Figure 2A-2F shows three example LIP neurons that exhibited significant motion coherence correlated DS. Surprisingly, LIP neurons showed greater DS in the CT condition than in the IT condition, even though the same motion stimuli were used in the same spatial location for both conditions. The averaged population activity showed this DS difference between CT and IT conditions for all four coherence levels (Figure 2G, 2H). During presentation of their preferred motion direction, LIP neurons showed significantly elevated activity in the CT relative to the IT at all coherence levels (Figure S1A, S1B, nested ANOVA: P(high) = 0.0326, F = 4.65; P(medium) = 0.0088, 142 F = 7.03; P(low) = 0.0076, F = 7.32; P(zero) = 0.0124, F = 6.4), and a trend toward lower activity to the nonpreferred direction for CT vs. IT (Figure S1C, S1D, nested ANOVA: P(high) = 0.0994, F = 2.75; P(medium) = 0.0649, F = 3.12; P(low) = 0.0311, F = 4.73; P(zero) = 0.0273, F = 4.96). Most of the LIP neurons (48 of 83) showed such opposing trends in activity modulation between the preferred and nonpreferred directions (Figure 2I). These results indicated a nonlinear modulation of saccade choice on motion DS in LIP, aligned precisely with the response property of each neuron. This is unlikely to be driven by a linear gain modulation of saccade direction selectivity. Receiver operating characteristic (ROC) analysis further confirmed significantly greater motion DS in the CT condition than in the IT condition (Figure 2J 148 and 2K; nested ANOVA: P(high) = 5.0e-4, F = 12.44; P(medium) = 9.53e-6, F = 20.91; P(low) = 9.33e-7, F 149 = 26.03; P(zero) = 2.56e-8, F = 34.3). Such DS differences were observed even before stimulus onset. Moreover, LIP neurons exhibited similar levels of mean activity between different saccade directions (CT vs. IT) before monkeys’ saccade choice (Figure 2L), further supporting that saccade direction selectivity did not significantly contribute to the observed modulation of LIP neurons’ responses to motion stimuli.

We also thank the reviewer for pointing out the missing of this relevant study, we have added the suggested refence in the revised discussion section as follows:

‘A recent study demonstrated that neurons in the middle temporal area responded more strongly to motion stimuli when monkeys saccaded toward their RFs in a standard decision task with a fixed mapping between motion stimuli and saccade directions. This modulation emerged through the training process and contributed causally to the monkeys' following saccade choices. Consistently, we found that the response of LIP neurons to motion stimuli was more strongly correlated with the monkeys' decisions in the CT condition (saccades toward RFs) than in the IT condition, in a more flexible decision task. Together, these results suggest that the modulation of action selection on sensory processing may be a general process in perceptual decision-making. However, the observed modulation of saccade direction on LIP neurons' responses to motion stimuli cannot be simply explained by saccade direction selectivity. Several lines of evidence argue against this possibility. First, the modulation effect was nonlinear; specifically, neuronal firing rates increased for preferred motion directions but decreased for non-preferred directions (Figure 2I and Figure S1). This pattern is unlikely to be driven by a linear gain modulation based on saccade directions. Second, we found that LIP neurons exhibited similar levels of activity in both the CT and IT conditions (Figure 2L), which is inconsistent with the presence of clear saccade direction selectivity.

Some more specific comments are below:- I had a bit of a hard time with the abstract. It does not appear to be crystal clear to me, and it is the first thing that I am reading after the title. For example, if there is a claim about both perceptual decision-making and later target selection, then I feel that the task should be explained a bit more clearly than saying "flexible decision" task. Also, "..modulated by monkeys' following saccade choices directing outside each neuron's response field" was hard to read. It needs to be rewritten. Maybe just say "...modulated by the subsequent eye movement choices, even when these eye movement choices always directed the eyes away from the recorded neuron's response field". Also, I don't fully understand what "selectivity-specific feedback" means. Then, the concept of "consistency" in flexible decisions is brought up, again without much context. The above are examples of why I had a hard time with the abstract.

We realize that our original statement may have been unclear and potentially caused confusion for the readers. Following the reviewer’s suggestions, we have revised the abstract as follows:

‘Neural activity in the primate brain correlates with both sensory evaluation and action selection aspects of decision-making. However, the intricate interaction between these distinct neural processes and their impact on decision behaviors remains unexplored. Here, we examined the interplay of these decision processes in posterior parietal cortex (PPC) when monkeys performed a flexible decision task, in which they chose between two color targets based on a visual motion stimulus. We found that the PPC activity related to monkeys’ abstract decisions about visual stimuli was nonlinearly modulated by their subsequent saccade choices, which were directed outside each neuron’s response field. Recurrent neural network modeling indicated that the feedback connections, matching the learned stimuli-response associations during the task, might mediate such feedback modulation. Further analysis on network dynamics revealed that selectivity-specific feedback connectivity intensified the attractor basins of population activity underlying saccade choices, thereby increasing the reliability of flexible decisions. These results highlight an iterative computation between different decision processes, mediated primarily by precise feedback connectivity, contributing to the optimization of flexible decision-making.’

Specifically, selectivity-specific feedback refers to the feedback connections with positive or negative weights between selectivity-matched and selectivity-nonmatched unit pairs, respectively.

Regarding "decision consistency," we define it as the degree to which the model’s decisions remain congruent under specific conditions. A higher level of decision consistency indicates that the model is more likely to produce the same choice each time it is presented with a stimulus under those conditions, in another words, decision reliability. We have revised the corresponding results section to make these concepts clearer.

- Line 69: I'm not fully sure, but I think that some people might suggest that superior colliculus is also involved in the sensory aspect of the evaluation. But, I guess the sentence itself is correct as you write it. So, I don't think anyone should argue with it. However, if someone does argue with it, then they would flag the next sentence, since if the colliculus does both, then do the sensory and motor parts really employ distinct neural processes? Anyway, I think this is very minor.

This is an interesting point. We have also noticed a recent study that demonstrates that the superior colliculus is causally involved in the sensory aspect of decision-making, specifically in visual categorization. However, the study also distinguishes between neural activity related to categorical decisions and that related to saccade planning. This suggests that the sensory and motor aspects of decision-making likely involve distinct neural processing, even within the same brain region—potentially reflecting separate populations of neurons. Therefore, we stand by our statement in the ‘next sentence’.

- Line 79-80: you might want to look at this work because I feel that it is relevant to cite here: https://www.biorxiv.org/content/10.1101/2023.08.03.551852v2

We have discussed this reference in the revised discussion section of the manuscript, please refer to the above response.

- For a result like that shown in Fig. 2, I feel that it is important to show RF mapping with a saccade task alone. i.e. for the same neurons, have a monkey make a delayed visually guided saccade task to all possible locations on the display, and demonstrate that there is no modulation by saccades to the targets. Otherwise, the result in Fig. 2 could reflect first an onset response by a motion, and then the saccade-related response that would happen anyway, even without the decision task. So, I feel that now, it is not entirely clear whether the result reflects this so-called feedback modulation, or whether simply planning the saccade to the target itself activates the neurons. With large RF's, this is a distinct possibility in my opinion.- Line 174: this would also be predicted if the neuron's were responding based on the saccade target plan independent of the motion stimulus- On a related note, I would recommend plotting all data also aligned on saccade onset. This can help establish what the cause of the effects described is

We understand the reviewer’s concern that the modulation might be related to saccade planning, and we acknowledge that the original manuscript might not adequately address this potential confound. Unfortunately, we did not map the LIP neurons' receptive fields (RFs) using a saccade-only task. However, as mentioned earlier, we believe that the modulation of LIP neurons' responses to motion stimuli based on saccade choice direction cannot be simply attributed to saccade direction selectivity. Several lines of evidence support this conclusion. First, the modulation we observed was nonlinear: the firing rate of neurons increased for the preferred motion direction but decreased for the non-preferred motion direction (Figure 2i and Figure S1A-D). This pattern is inconsistent with a simple linear gain modulation driven by saccade direction selectivity. Second, we directly compared LIP neuronal activity for contralateral and ipsilateral target conditions, and found no significant differences between the two. This suggests that saccade direction selectivity is unlikely to be the primary contributor to the observed modulation. In the revised figure, we added a plot (Figure 2L) that aligns neural activity to saccade onset, in addition to the original alignment to motion stimulus onset (Figure S1E). This new analysis further supports our interpretation.

**Author response image 7. sa2fig7:** 

- Even when reading the simulation results, I'm still not 100% sure I understand what is meant by this idea of "consistency" of flexible decision-making

We have addressed this issue in a previous comment and please refer to the response above.